# Flat Channels to Infinity in Neural Loss Landscapes

**Flavio Martinelli**[1]* **Alexander van Meegen**[1]* **Berfin Şimşek**[2]
**Wulfram Gerstner**[1] **Johanni Brea**[1]
[1] EPFL, Lausanne, Switzerland
[2] Flatiron Institute, New York, USA
*equal contribution
{flavio.martinelli,alexander.vanmeegen,johanni.brea}@epfl.ch

## Abstract

The loss landscapes of neural networks contain minima and saddle points that may be connected in flat regions or appear in isolation. We identify and characterize a special structure in the loss landscape: channels along which the loss decreases extremely slowly, while the output weights of at least two neurons, $a_i$ and $a_j$, diverge to $\pm$infinity, and their input weight vectors, $\boldsymbol{w}_i$ and $\boldsymbol{w}_j$, become equal to each other. At convergence, the two neurons implement a gated linear unit: $a_i\sigma(\boldsymbol{w}_i \cdot \boldsymbol{x}) + a_j\sigma(\boldsymbol{w}_j \cdot \boldsymbol{x}) \to c\sigma(\boldsymbol{w} \cdot \boldsymbol{x}) + (\boldsymbol{v} \cdot \boldsymbol{x})\sigma'(\boldsymbol{w} \cdot \boldsymbol{x})$. Geometrically, these channels to infinity are asymptotically parallel to symmetry-induced lines of critical points. Gradient flow solvers, and related optimization methods like SGD or ADAM, reach the channels with high probability in diverse regression settings, but without careful inspection they look like flat local minima with finite parameter values. Our characterization provides a comprehensive picture of these quasi-flat regions in terms of gradient dynamics, geometry, and functional interpretation. The emergence of gated linear units at the end of the channels highlights a surprising aspect of the computational capabilities of fully connected layers.

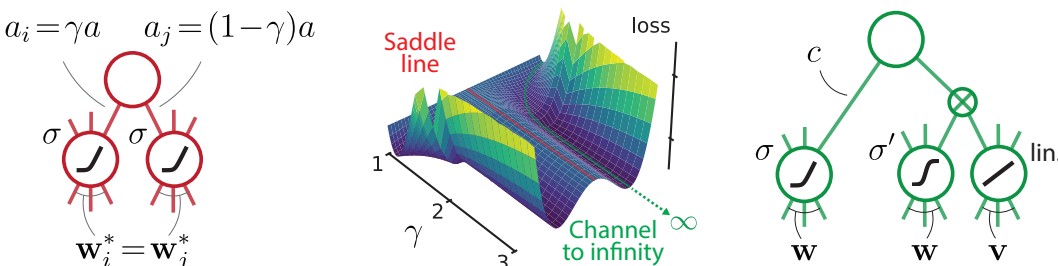

Figure 1: **Saddle lines à la Fukumizu & Amari [1] and channels to infinity.** *Left*: Duplicating a neuron in a network trained to convergence generates lines of saddle points in the loss landscape [1]. Duplicated neurons share the input weights of the original neuron while their output weights $\gamma a, (1 - \gamma)a$ sum to the original neuron's output weight $a$. *Middle*: Loss landscape of duplicated network projected along the saddle line (in red) and the eigenvector of the smallest (most negative) eigenvalue of the loss Hessian. Parallel to the saddle line there are channels to infinity (green curve) along which the loss decreases very slowly. Following the channel, the output weights diverge to infinite norm and the input weights converge to a new value. *Right*: The solution at infinity implements a new function consisting of a single neuron and a *gated linear unit*. The gating function is the derivative of the original activation function $\sigma$.

39th Conference on Neural Information Processing Systems (NeurIPS 2025).

# 1 Introduction

Although neural network loss functions are known to be non-convex, machine learning practitioners usually find good solutions with gradient-descent-like methods. This is puzzling, because the non-convexity is not obviously benign [2, 3, 4]. One simple example of non-convexity are critical points resulting from embedded sub-networks. In fact, an arbitrarily large network can be constructed by duplicating hidden neurons of smaller networks, such that the optimal loss of any embedded sub-network appears as a manifold of critical points in the loss function of the full network [1, 5, 6]. Typically, neuron duplication results in the transformation of local minima into a line of saddle points [5, 6], but it is also possible that this line contains segments of local minima [1, 5]. There are good theoretical arguments that local minima are unlikely, in particular in multilayer settings (see Appendix A and [7]), and that the lines of saddle points are unproblematic [8]. Thus, the saddle lines are expected to be a benign non-convexity. Empirically, we confirm that these symmetry-induced saddle lines are unproblematic even in low-dimensional settings (see section 2 and Appendix A). However, we find that there are many channels in the loss landscape that are asymptotically parallel to symmetry-induced lines of saddle points, and lead to local minima at infinity (see Figure 1). These minima at infinity differ from the well-known solutions at infinity of separable classification problems [9]. They are characterized by input vectors of at least two neurons becoming equal to each other at finite values, while their output weights diverge to $\pm$ infinity. Our main contributions are:

1. We identify and characterize a novel structure in the loss landscape of fully connected layers: the channels to infinity (section 3).
2. We show empirically that gradient-based training reaches these channels with high probability from the random initialization schemes used in practice (section 3).
3. We explain the functional role of the channels to infinity: they endow the network with novel computational capabilities by implementing a gated linear unit (section 4).

## 1.1 Related Work

*Neuron splitting creates a continuum of critical points:* Fukumizu and Amari [1] were the first to report the mapping of critical points of the loss function of a neural network to a *line* of critical points of the loss function of a new neural network with an extra neuron. Moreover, they studied the Hessian of the loss on this line of critical points and showed that regions of local minima appear, if a certain condition is satisfied. When this condition is not satisfied, neuron splitting creates strict saddles enabling escape [10, 11], which is utilized to propose alternative algorithms to train neural networks by sequentially growing them [12, 13]. With similar reasoning, splitting of multiple neurons [5], and neuron-splitting in deep networks [14, 7] were analyzed.

*Sub-optimal minima:* For extremely wide neural networks, the loss function does not possess any sub-optimal minimum under general conditions [15, 16, 17] (see [18] for a review). However, there are families of sub-optimal minima for non-overparameterized and mildly-overparameterized neural networks [19, 20, 21]. Neuron splitting turns sub-optimal minima into saddles [1, 5, 7], and an overparameterization with a few extra neurons makes the optimization problem significantly easier [22, 23]. In the context of the spiked tensor model, an intricate structure of saddle points allows gradient flow to circumvent sub-optimal local minima [24].

*Flat regions in loss-landscapes:* Precisely-flat regions in loss landscapes are described in various works, from manifolds of symmetry-induced critical points [1, 5, 6, 10] to manifolds of global minima in overparameterized [25, 26, 27, 28, 29, 30] and teacher-student setups [6, 23]. Empirically, flat regions in loss landscapes were found in interpolations between solutions of different training runs, a phenomenon typically called mode-connectivity [31, 32], more prominently after permutation alignment [33, 34]. These locally flat manifolds have been discovered to be star-shaped [35, 36] or to present orthogonal tunnels that spread in the parameter norm [37].

*Attractors at infinity in neural networks:* The well-known minima at infinity in separable classification problems [9] differ from the ones we investigate here. In linear networks, and in one-dimensional sigmoidal deep networks trained with a *single* datapoint, local minima at infinity do not occur [38], but there are examples where minima at infinity can arise with two or more datapoints [38, 39]. It is possible to eliminate finite-norm local minima from landscapes through the addition of specific neurons directly connecting input to output [40, 41], and eliminating local minima at infinity with regularization [42].

## 1.2 Setup

To expose the key ideas, we consider neural networks with a single hidden layer and mean-squared error loss. The main insights generalize to multiple layers and do not depend on the loss function (see section 2). More explicitly, we consider the loss landscape

$$\mathcal{L}_r(\boldsymbol{\theta}; \mathcal{D}) = \frac{1}{N} \sum_{i=1}^{N} \left( y^{(i)} - \sum_{j=1}^{r} a_j \sigma\big(\boldsymbol{w}_j \cdot \boldsymbol{x}^{(i)}\big) \right)^2. \tag{1}$$

The parameters are $\boldsymbol{\theta} = (\boldsymbol{w}_1, \ldots, \boldsymbol{w}_r, a_1, \ldots, a_r)$, the data $\mathcal{D} = \big\{(\boldsymbol{x}^{(i)}, y^{(i)}))\big\}_{i=1}^{N}$ is fixed, and $\sigma$ is a continuous activation function. The input samples $\boldsymbol{x}^{(i)} \in \mathbb{R}^{d+1}$ are composed of $d$-dimensional input vectors and a constant $x_{d+1}^{(i)} = 1$ to include a bias term $w_{d+1}$ in $\boldsymbol{w} \cdot \boldsymbol{x} = \sum_{k=1}^{d} w_k x_k + w_{d+1}$.

## 2   Neuron duplication introduces lines of critical points

We start with an empirical investigation of the symmetry-induced saddle lines. Fukumizu and Amari ([1], Theorem 1) showed that any critical point of the loss function with $r$ neurons implies an equal-loss line of critical points of the loss function with $r + 1$ neurons. Formally, if $\boldsymbol{\theta}^* = (\boldsymbol{w}_1^*, \ldots, \boldsymbol{w}_r^*, a_1^*, \ldots, a_r^*)$ is a critical point of the loss function, i.e., $\nabla_{\boldsymbol{\theta}} \mathcal{L}_r(\boldsymbol{\theta}^*; \mathcal{D}) = \mathbf{0}$, then the parameters

$$\boldsymbol{\theta}^{\gamma} = (\underbrace{\boldsymbol{w}_1^*, \ldots, \boldsymbol{w}_r^*, \boldsymbol{w}_r^*}_{r+1 \text{ vectors}}, \underbrace{a_1^*, \ldots, a_r^*, 0}_{r+1 \text{ weights}}) + \gamma a_r^*(\underbrace{\mathbf{0}, \ldots, \mathbf{0}, \mathbf{0}}_{r+1 \text{ vectors}}, \underbrace{0, \ldots, -1, +1}_{r+1 \text{ weights}}) \tag{2}$$

of a neural network with one additional neuron are also at a critical point, i.e., $\nabla_{\boldsymbol{\theta}} \mathcal{L}_{r+1}(\boldsymbol{\theta}^{\gamma}; \mathcal{D}) = \mathbf{0}$ for any $\gamma \in \mathbb{R}$, and $\mathcal{L}_r(\boldsymbol{\theta}^*; \mathcal{D}) = \mathcal{L}_{r+1}(\boldsymbol{\theta}^{\gamma}; \mathcal{D})$. The variable $\gamma$ parametrizes the line $\Gamma = \{\boldsymbol{\theta}^{\gamma} : \gamma \in \mathbb{R}\}$ that points in direction $(\mathbf{0}, \ldots, \mathbf{0}, 0, \ldots, +1, -1)$; we call this line the *saddle line*. The stability of the symmetry-induced critical points $\boldsymbol{\theta}^{\gamma}$ of the $\mathcal{L}_{r+1}$ loss depends on the specific choice of $\gamma$, and the spectrum of a symmetric $(d+1) \times (d+1)$-dimensional matrix ([1]; Theorem 3). If and only if this matrix is positive or negative definite, there is a region of local minima on the saddle line, which we call a *plateau saddle*, because it is bounded by strict saddle points (see Figure 2c).

Given the amount of available duplications in a network, the number of saddle lines in the loss landscape grows factorially with network width [6]. What are the chances of finding a stable region of the saddle line – a plateau saddle – in the loss landscape? To obtain a comprehensive view of all minima, we trained an extensive set of small networks of increasing widths on a $d = 2$, scalar regression problem (Figure 2, Appendix A). In a setup where neurons have no bias and the output is a scalar, we find that many saddle lines contain plateau saddles, where gradient dynamics converge. This is evident in Figure 2b, where networks of different sizes converge to identical loss values, since they compute identical functions. These converged networks contain at least two neurons of equal input weight vector (Equation 2), the signature of a plateau saddle. Convergence on a plateau saddle occurs with probabilities from 10% to 30% across random initializations (Figure 2b, inset).

The loss landscape around plateau saddles is studied in Figure 2d, where we perform a perturbation analysis on a solution found in Figure 2, and analyze its stability. Plateau saddles are attracting (all eigenvalues of the Hessian are non-negative), flat, regions of the landscape (Figure 2c), they can be present in the segment $\gamma \in (0, 1)$, or outside, or nowhere ([1]; Theorem 3). This distinction between segments of the saddle line is important, because at its boundaries ($\gamma = 0, 1$), some eigenvalues of the Hessian meet at zero, and swap sign; a signature of a highly degenerate point in the landscape. In all other regions of $\gamma$, eigenvalues do not cross and instead exhibit an eigenvalue repulsion phenomenon [43] (see Figure A4).

Since the number of unique minima can increase exponentially with network width, it is unfeasible to comprehensively characterize the set of minima for larger networks [24]. However, we verify empirically (Appendix A, Figure A1), that saddle lines lose almost all plateau-saddles if we add biases to the network. A similar conclusion was predicted with theoretical arguments for the case of multi-output or multi-layer MLPs [7]. Although these saddle lines do not seem to pose directly any problem in realistic settings, these equal loss lines through the entire parameter space may have a special impact on regions nearby.

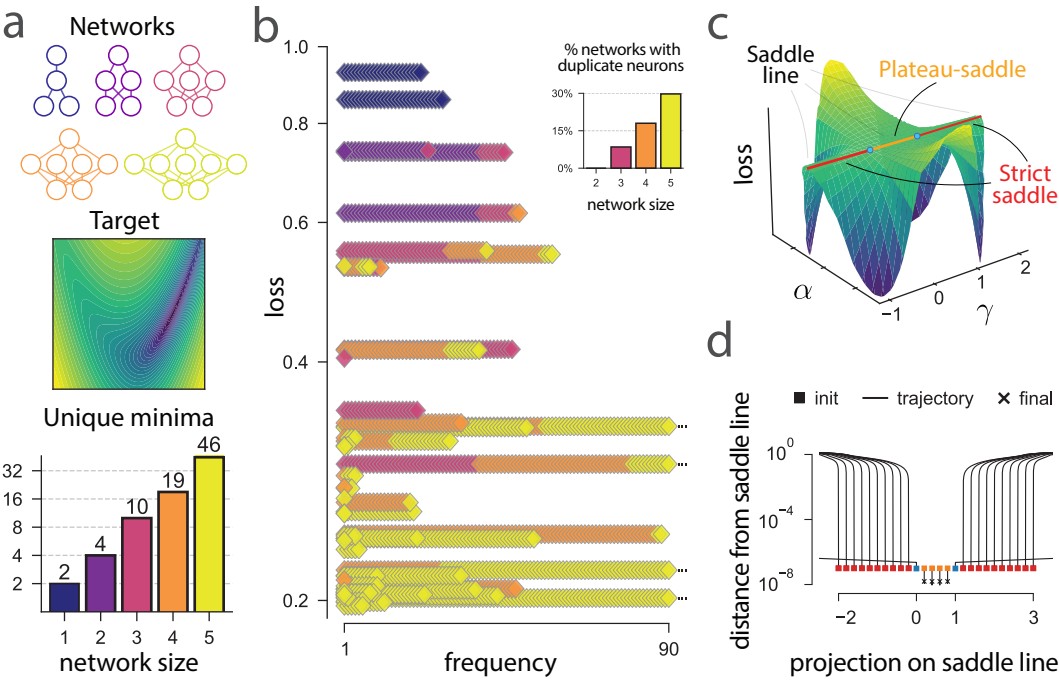

Figure 2: **Stable plateau-saddles and their loss landscape in MLPs without bias.** (a) Networks of 1 to 5 hidden neurons and scalar output are trained on the shown 2D regression target (logarithm of the rosenbrock function, see Appendix A). Training follows full-batch gradient flow dynamics until convergence to a critical point. A quantification of unique solutions in weight-space (up to permutation symmetries) is shown at the bottom. (b) Loss levels of converged networks: each diamond shows the loss of a converged network, color-code indicates network size. The only source of randomness is the initialization. Many identical-loss solutions are found by networks of different sizes. Inset: Frequency of converged solutions exhibiting duplicated neurons. (c) Loss landscape along the duplication parameter $\gamma$ and the direction of smallest eigenvalue of the Hessian $\alpha e_{\min}(\gamma)$ corresponding to one of the converged solutions shown in b. Small perturbations are stable only within the plateau-saddle region, $\gamma \in (0, 1)$. (d) Gradient-flow trajectories following a small perturbation from the saddle line in the direction of $\alpha e_{\min}$ for the example shown in c. Perturbations outside the plateau-saddle region, $\gamma \notin (0, 1)$, escape the saddle line and land in other minima.

## 3 Seemingly flat regions in the loss landscape as channels to infinity

Where do gradient descent dynamics converge, if initialized near a saddle line? Using ODE solvers to integrate the gradient flow dynamics $\dot{\theta} = -\nabla_\theta \mathcal{L}_r(\theta; \mathcal{D})$ [44], we discovered *channels* in the loss landscape that are characterized by:

1. diverging output weights of at least two neurons, $|a_i| + |a_j| \to \infty$ of opposed signs $a_i a_j < 0$,
2. decreasing distance between corresponding input weight vectors $d(w_i, w_j) \to 0$.

Due to these properties we refer to these structures as *channels to infinity*.

Figure 3 shows a detailed example of the landscape around a saddle line. To first produce a saddle line we (i) train an MLP until convergence into a finite-norm local minimum $\theta$, (ii) apply neuron-duplication, $\theta \to \theta^\gamma$, at various $\gamma$ values (Equation 2), and (iii) compute the loss function on a planar slice spanning $\theta^\gamma$ and the vector $\alpha e_{\min}$, where $\alpha \in \mathbb{R}$ and $e_{\min}$ is the smallest (most negative) eigenvalue of the loss Hessian, $\nabla_\theta^2 \mathcal{L}_{r+1}(\theta^\gamma; \mathcal{D})$. In Figure 3a, we see the landscape characterized by a channel of decreasing loss running parallel to the saddle line. These channels continue indefinitely in the direction of the saddle line, Figure 3c. Since this view is neglecting all other dimensions of the loss landscape, we conducted a perturbation analysis from the saddle line to track the gradient-flow trajectories. In this specific example, only one of the two channels is attractive. Trajectories of dynamics trapped in the channel are characterized by a very slow decrease in loss (Figure 3d), and

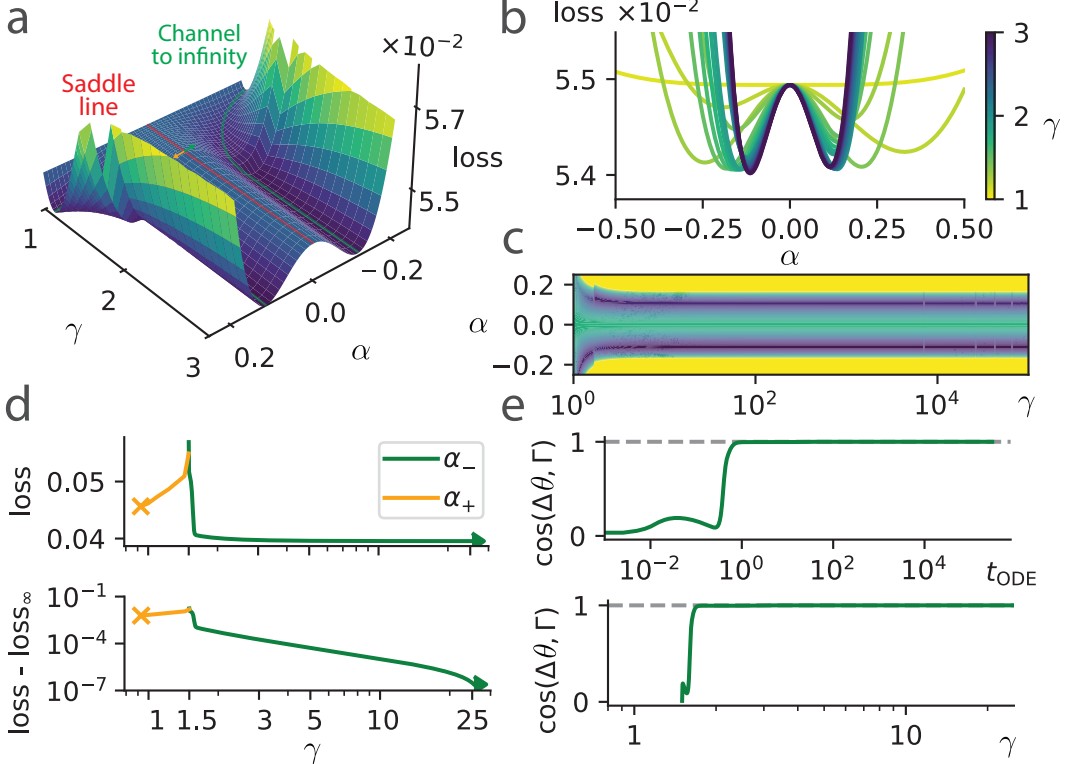

Figure 3: **Channels to infinity.** (a) Loss landscape of a 4-4-1 MLP trained on a regression task (Appendix B). The saddle line (red straight line) is found via neuron splitting from a local minimum of a 4-3-1 MLP. The surface is a slice of the loss along the splitting parameter $\gamma$ and the direction of smallest (negative) eigenvalue of the Hessian $\alpha e_{\min}$. Most other eigenvalues are positive. At first glance, it looks as if there were two channels to infinity parallel to the saddle line, but the analysis in the next panels reveals that there is only one (the green curved line). (b) Loss profile along $\alpha e_{\min}$, color-coded at different values of $\gamma$. Note that the loss is not continually decreasing for positive $\alpha$, indicating that this is not a channel to infinity. (c) A top-view of the landscape for large $\gamma$ reveals that the local picture of the loss landscape in (a) holds also for very large $\gamma$. (d) The two-dimensional projection of the loss landscape in panels (a)-(c) does not show how the loss depends on all the other free parameters of the 4-4-1 MLP. Therefore, we look at gradient-flow trajectories following a small perturbation along $e_{\min}$ from the saddle line at $\gamma = 1.5$. The perturbation direction is shown as green and orange arrows on the surface plot in panel (a). After the green perturbation ($\alpha_-$), the gradient trajectory moves inside a channel to infinity towards increasing values of $\gamma$ following a descent with extremely small slope (green channel to infinity). The orange perturbation ($\alpha_+$) converges to a finite-norm minimum, which confirms that the landscape at positive $\alpha$ is not a channel to infinity. (e) Cosine distance between parameter updates $\Delta\theta$ and direction of the saddle line ($\Gamma$) for the $\alpha_-$ perturbation: after an initial high-dimensional trajectory, parameter updates $\Delta\theta$ are parallel to the saddle line. The ODE dynamics reveal an extremely slow divergence of $\gamma \to \infty$.

updates that become parallel to the saddle line (Figure 3e). To report $\gamma$ for a trajectory that is not on a saddle line, we measure the projected $\gamma$ by averaging the $\gamma$ estimates $a_r/a_r^*$ (for neuron $r$), and $(a_r^* - a_{r+1})/a_r^*$ (for neuron $r + 1$), leading to $\gamma = (a_r - a_{r+1} + a_r^*)/2a_r^*$, where $a_r^*$ is the output weight of the original duplicated neuron and $a_r, a_{r+1}$ are the output weights after duplication. Note that $\gamma \to \pm\infty \Rightarrow |a_r| + |a_{r+1}| \to \infty$.

Figure 4 shows summary statistics of channels to infinity found across different training conditions; importantly, we start in all cases from random initial conditions (Glorot normal initialization). We trained MLPs with different architectures – varying layer sizes, input dimensionality, shallow and deep MLPs – and across various datasets – the modified rosenbrock function (a log-polynomial) and four types of Gaussian processes (GP) characterized by different kernel scalings $s \in 0.1, 0.5, 2, 10$ (see Appendix B for more details). All experiments were performed with biases and softplus activation

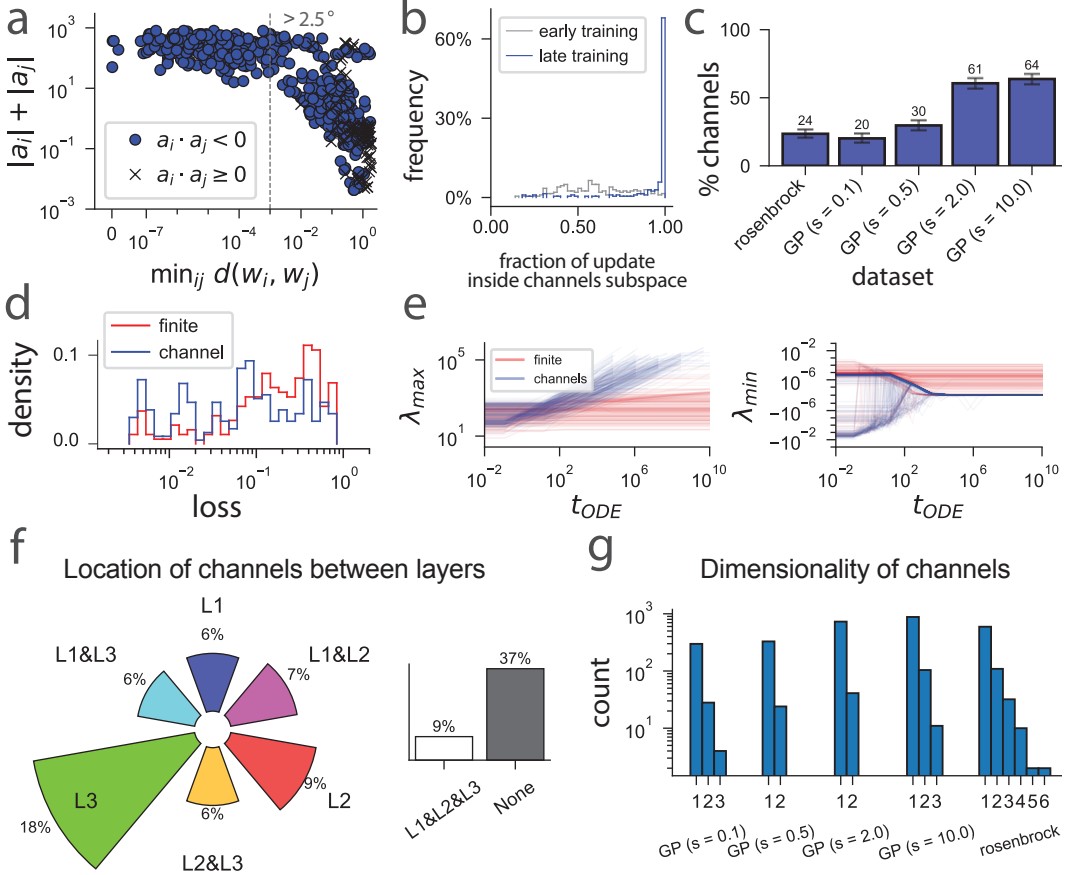

Figure 4: **Frequency and properties of channels to infinity.** (a) As a first criterion to identify channels to infinity, we consider the cosine distance of the pair of closest input weight vectors within a network and the sum of absolute output weights corresponding to that pair. Putative channel solutions are identified by having a large weight norm and a small distance in input weights (top left section of the graph). (b) As a second criterion, we consider channel solutions networks that had parameter updates mostly within the subspace spanned by the putative channels, parallel to the saddle line. After filtering for networks that satisfy (a), we compute the percentage of updates that lie inside the channels subspace. At late stage of training, most of the network updates are parallel to the saddle line (c.f. Figure 3e). (c) Estimated probability of converging to a channel for various datasets and architectures spanning different input and hidden dimensions and number of hidden layers (see Appendix B for details). (d) Distribution of finite-norm minima and channel minima training loss. There is no evident difference between types of minima (see Appendix B). (e) Trajectories of maximum and minimum Hessian eigenvalues for finite-norm and channel minima. Channel solutions have larger maximum eigenvalues, indicating that they are sharper than the finite-norm minima. Channels sharpen as training progresses. They are extremely flat regions, as indicated by the small magnitude of the minimum eigenvalue. (f) For MLPs with three hidden layers, channels appear in all layers, and in multiple layers at the same time. (g) Channels do not always involve pairs of neurons, they can be formed by an arbitrary number of neurons with diverging output weights and converging input weights. In these cases, the flat regions are multi-dimensional (see Appendix B for details on multi-dimensional channels). (b-c-d-e) show results for the GP (s=0.5) dataset; see Appendix B for other datasets.

function. We identify channels following three criteria: (i) high parameter norm, (ii) low distance between at least a pair of input weight vectors, (iii) updates only within the parameters contributing to the channels (more details in Figure 4a,b, and Appendix B). We observe a substantial probability of reaching channels in all tested dataset. More channels are found when the target function is rough, as evidenced by the positive correlation with the scale of the GP kernels $s$ (Figure 4c, Figure B8). The distribution of losses between channels and finite-norm local minima seem to be similar (Figure 4d),

suggesting that these solutions may not be worse than other local minima. Maximum and minimum eigenvalue of the Hessian for all networks trained on the GP($s = 0.5$) dataset are shown in Figure 4e. We observe a clear difference between finite-norm solutions (red) and channels: as training progresses the channels become both sharper and flatter than finite-norm minima (see Figure B12 for a slice of the landscape along the steepest direction). In multilayer MLPs, channels can appear in any layer, and also simultaneously in different layers (Figure 4f). Finally, channels are not limited to be one-dimensional: multiple pairs of neurons can generate higher dimensional channels to infinity (Figure 4g). We tested networks with smooth activation functions, such as softplus, erf, sigmoid, and found channels appearing in all settings (see Appendix B for further details, including a brief discussion of relu activation functions).

In summary, striking features of these channels are their extreme flatness along the direction of the saddle line, and the increasing sharpness as the parameter norm diverges to infinity. As the gradient flow dynamics does not seem to stop in these channels, we hypothesize that they lead to local minima at infinity.

## 4 Channels to infinity converge to gated linear units

What is the functional role of the channels? To better understand the convergence in these channels, we consider two neurons $i$ and $j$ and reparameterize them as

$$
a_i\sigma(\boldsymbol{w}_i \cdot \boldsymbol{x}) + a_j\sigma(\boldsymbol{w}_j \cdot \boldsymbol{x}) = \frac{c}{2}\left(\sigma\big((\boldsymbol{w} + \epsilon\boldsymbol{\Delta}) \cdot \boldsymbol{x}\big) + \sigma\big((\boldsymbol{w} - \epsilon\boldsymbol{\Delta}) \cdot \boldsymbol{x}\big)\right)
$$
$$
+ \frac{a}{2\epsilon}\left(\sigma\big((\boldsymbol{w} + \epsilon\boldsymbol{\Delta}) \cdot \boldsymbol{x}\big) - \sigma\big((\boldsymbol{w} - \epsilon\boldsymbol{\Delta}) \cdot \boldsymbol{x}\big)\right) \tag{3}
$$

with $\boldsymbol{w} = (\boldsymbol{w}_i + \boldsymbol{w}_j)/2, \epsilon = \|\boldsymbol{w}_i - \boldsymbol{w}_j\|/2, \boldsymbol{\Delta} = \frac{\boldsymbol{w}_i - \boldsymbol{w}_j}{\|\boldsymbol{w}_i - \boldsymbol{w}_j\|}, a = \epsilon(a_i - a_j), c = a_i + a_j$. The second term in Equation 3 is the central finite difference approximation of the derivative in direction $\boldsymbol{\Delta}$. In the limit $\epsilon \to 0$ with fixed $\boldsymbol{w}, \boldsymbol{\Delta}, a, c$, the central difference converges to the derivative,

$$
a_i\sigma(\boldsymbol{w}_i \cdot \boldsymbol{x}) + a_j\sigma(\boldsymbol{w}_j \cdot \boldsymbol{x}) \xrightarrow{\epsilon \to 0} c\sigma(\boldsymbol{w} \cdot \boldsymbol{x}) + a(\boldsymbol{\Delta} \cdot \boldsymbol{x})\sigma'(\boldsymbol{w} \cdot \boldsymbol{x}). \tag{4}
$$

Interestingly, the second term implements a gated linear unit, with gate $\sigma'(\boldsymbol{w} \cdot \boldsymbol{x})$ and linear transformation $\boldsymbol{v} \cdot \boldsymbol{x} = a(\boldsymbol{\Delta} \cdot \boldsymbol{x})$. If $\sigma(x) = \log(1 + \exp(x))$ is the softplus activation function, the gate is given by the standard sigmoid $\sigma'(x) = 1/(1 + \exp(-x))$ used in GLU [45]. Importantly, the limit $\epsilon \to 0$ with fixed $a$ corresponds exactly to converging input weights ($\|\boldsymbol{w}_i - \boldsymbol{w}_j\| = 2\epsilon$), and diverging output weights $a_i - a_j = a/\epsilon$, which we observed in many solutions (Figure 4). Note that the limit $\epsilon \to 0$ with fixed $a$ implies that the readout weights diverge with $1/\epsilon$.

To clarify the geometric relation between the channels and the saddle line we write the above reparameterization in the full parameter space as

$$
\boldsymbol{\theta} = (\boldsymbol{w}_1, \dots, \boldsymbol{w}, \boldsymbol{w}, a_1, \dots, c/2, c/2) + (\boldsymbol{0}, \dots, \epsilon\boldsymbol{\Delta}, -\epsilon\boldsymbol{\Delta}, 0, \dots, a/2\epsilon, -a/2\epsilon) \tag{5}
$$

where we chose neurons $i$ and $j$ to be the last two neurons w.l.o.g. For small $\epsilon$ the second term becomes $\frac{a}{2\epsilon}(\boldsymbol{0}, \dots, \boldsymbol{0}, \boldsymbol{0}, 0, \dots, 1, -1)$ which is parallel to the second term in Equation 2 – the channels become asymptotically parallel to the saddle line with $\gamma = 1/\epsilon$.

A priori, it is unclear whether the loss function in Equation 1 has minima at the end of channels to infinity; it could be that the optimal $\epsilon$ in Equation 3 is at a finite value. Empirically, we never saw the ODE solvers stop in the channels, but they make rather slow progress, and with reasonable compute budgets we never reached values of $1/\epsilon$ above a few hundreds, even with small networks. To obtain further empirical evidence for the optimum at infinity, we used a jump procedure, which alternates between dividing $\epsilon$ by a constant factor, and integrating the gradient flow for a mixed amount of time, to move quickly along the channels (see Appendix C). With this jump procedure we see that the loss continually decreases, $c, a, \boldsymbol{w}, \boldsymbol{\Delta}$ seem to converge, and the approximation error in Equation 4 decreases quadratically with $\epsilon$ (see Figure 5).

Proving convergence to the gated linear unit in Equation 4 is difficult for general settings. However, expanding the loss in $\epsilon$ around $\epsilon = 0$, leads to $\mathcal{L}_r(\boldsymbol{\theta}; \mathcal{D}) = \mathcal{L}_r(\boldsymbol{\theta}_0; \mathcal{D}) + \epsilon^2 h(\boldsymbol{\theta}_0; \mathcal{D}) + \mathcal{O}(\epsilon^4)$, where $\boldsymbol{\theta}_0$ are the parameters for $\epsilon \to 0$, and $h(\boldsymbol{\theta}_0; \mathcal{D})$ denotes the leading order correction term. We see this scaling of the loss with $\epsilon^2$ in Figure 5. If $\boldsymbol{\theta}_0$ is a local minimum of the loss, it is attractive in $\epsilon$, if $h(\boldsymbol{\theta}_0; \mathcal{D}) > 0$ (to prove full stability, one would need to look at the spectrum of the entire Hessian at

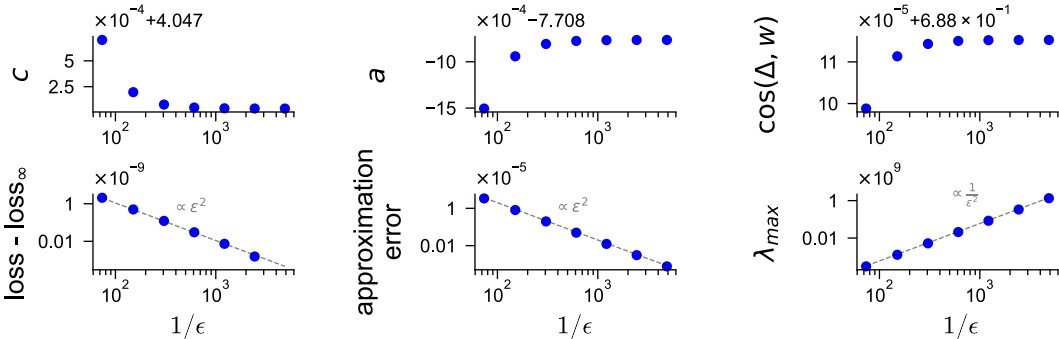

Figure 5: **Convergence in $\epsilon$ to gated linear units.** Moving along a channel to infinity with the jump procedure described in Appendix C shows that $c$, $a$, and the cosine similarity $\cos(\boldsymbol{\Delta}, \boldsymbol{w})$ converge to constant values, and that the loss and the approximation error decrease with $\epsilon^2$ and the sharpness diverges with $1/\epsilon^2$, as predicted by the theory. For this example, a network with 8 input dimensions and 8 hidden softplus neurons (81 parameters) trained on the rosenbrock target function was used.

$\boldsymbol{\theta}_0$). This scaling also characterizes the asymptotic flatness along the channel: combining $d\mathcal{L}_r \sim \epsilon d\epsilon$ with $d(a_i - a_j) \sim -\epsilon^{-2}d\epsilon$ leads to $d\mathcal{L}_r/d(a_i - a_j) \sim -\epsilon^3$, i.e., the slope decreases asymptotically with $\epsilon^3$ along the channel.

To gain further insights, we turned to a toy example, where a network with two hidden neurons without biases (activation function $\sigma(x) = \mathrm{erf}(x/\sqrt{2})$) is used to fit the one-dimensional target function $f(x) = \mathrm{erf}\left((5x + 2.5)/\sqrt{2}\right) + \mathrm{erf}\left((5x - 2.5)/\sqrt{2}\right)$ (see Figure 6a). This network has only 4 parameters: $a_1, a_2, w_1, w_2$ or, equivalently, $c, a, w, \epsilon$ ($\Delta = 1$, in one input dimension). Importantly, the wiggle of the teacher function at $x = 0$ cannot be well approximated by a single $\mathrm{erf}$ function, but the combination of $\mathrm{erf}$ and its derivative $\mathrm{erf}'$ leads to an accurate approximation (Figure 6a). To avoid finite data effects, we base the analysis on the population (or infinite-data) loss

$$\ell(a_1, a_2, w_1, w_2) = \int_{-\infty}^{\infty} \left(f(x) - a_1\sigma(w_1 x) - a_2\sigma(w_2 x)\right)^2 p(x)dx, \qquad (6)$$

where $p(x)$ is the standard normal density function. We can express the loss function in terms of normal integrals, find the optimal $c^*(w, \epsilon)$ and $a^*(w, \epsilon)$ analytically for any value of $w$ and $\epsilon$, and expand the loss $\ell(w, \epsilon) = \ell\left(c^*(w, \epsilon), a^*(w, \epsilon), w, \epsilon\right)$ around $\epsilon = 0$, to find $\ell(w, \epsilon) = \ell(w, 0) + \epsilon^2 h(w) + \mathcal{O}(\epsilon^4)$ (see Appendix C for $h(w)$ and the derivation). We find that the loss $\ell(w, 0)$ has three critical points for positive $w$ (the loss is symmetric $\ell(w, 0) = \ell(-w, 0)$): two correspond to local minima, and one to a saddle point of the full loss $\ell(c, a, w, \epsilon)$ (see Figure 6c). At the local minima $w_1^*$ and $w_2^*$ we find $h(w_1^*) > 0$ and $h(w_2^*) > 0$, which shows that the loss function has indeed stable fixed points at $\epsilon = 0$, or, equivalently, at diverging $a_1$ and $a_2$. We emphasize that in this example the optimal solution is at a local minimum at infinity (at the end of channel 2 in Figure 6b); we note that this is consistent with Proposition 1 in [39]. Note that a small weight regularization would prevent finding the optimal solution due to the flatness of the channel.

In this toy example, we can also see that stochastic gradient descent (SGD) or ADAM [46] easily finds the "entrance" of the channels to infinity (Figure 6b), but quickly gets stuck there. This occurs because the trajectory reaches the 'edge of stability' [47]: along the channel, the sharpness (maximum eigenvalue of the Hessian) increases until it reaches the maximal value $\lambda_{\max} = 2/\eta$ for a given step size $\eta$ (Figure 6d). Since the Hessian contains terms that scale with $a_i a_j \sim 1/\epsilon^2$, its largest eigenvalue diverges with $\lambda_{\max} \sim 1/\epsilon^2$ along the channel, leading to $\epsilon_{\min} \sim \sqrt{\eta}$. Curiously, the increasing sharpness coexists (in orthogonal dimensions) with the increasing flatness of the channel (compare Figure 3c and Figure B12).

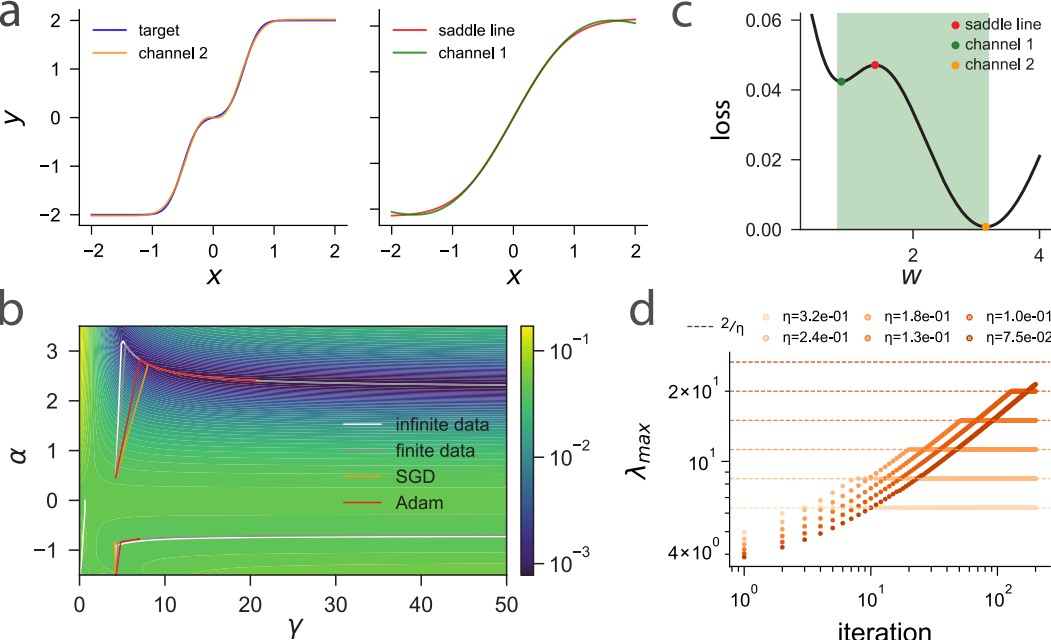

Figure 6: **Minima at infinity and edge of stability.** (a) The solution at infinity (channel 2) can be the global minimum, as in this example, where a one-dimensional target function (blue curve) is fitted with a neural network with two $\mathrm{erf}$ neurons without biases (4 parameters). (b) The loss landscape has three stable solutions: (i) the plateau saddle for $\gamma \in (0, 1)$ (white line on the left converging to $\alpha = 0, \gamma = 0.65$), (ii) channel 1 to infinity for negative $\alpha$, and (iii) low-loss channel 2 to infinity for positive $\alpha$. Gradient flow with finite data follows closely gradient flow with infinite data; SGD and ADAM get stuck early in the channels. (c) In the limit $\epsilon \to 0$ we can express the loss $\ell(w, 0)$ as a function of $w$ (see section 4), which shows clearly that the channel solutions are stable in $w$. Additionally, in the green region, the function $h(w)$ in the $\epsilon$-expansion of the loss is positive, showing that the local minima are indeed reached in the infinite time limit. (d) Using GD with a finite step size $\eta$, the optimization cannot go beyond the 'edge of stability' where the maximum eigenvalue of the Hessian equals $\lambda_{\mathrm{max}} = 2/\eta$ (dashed lines; color indicates learning rate $\eta$).

# 5   Discussion

Neural network loss landscapes are non-convex, but the empirical success in training shows that their structure is benign also outside the deeply over-parameterized regime. We uncovered a generic mechanism contributing to this benign nature of the landscape: extremely flat *channels to infinity*. While these channels lead to local minima, they endow the network with new computational capabilities in the form of *gated linear units*, that arise as the limit of the central finite difference approximation of a directional derivative. Thus, the channels can give rise to good local minima. Channels involving multiple neurons can lead to yet another set of capabilities based on higher-order derivatives (see Appendix B), but a full characterization of these multi-neuron channels is an interesting topic of future research.

We developed our theory for the stylized setup of a single-hidden-layer MLP, mean-squared error loss, and smooth activation functions. However, the channels to infinity arise from a simple interaction of input and output weights of pairs of neurons. This mechanism neither depends on the type of loss nor the number of layers. The pair of neurons contributing to the channel can be located in an arbitrary fully connected layer of a deep network; indeed, this is the case empirically (Figure 4f). We expect that channels to infinity arise also in large-scale models, within MLP layers of transformer or convolutional architectures, and even between channels of convolutional layers, but a quantification of the occurrence probability of channels to infinity in these large scale settings is ongoing work. Furthermore, groups of similar neurons, corresponding to condition 2 of our channel identification (Appendix B), have already been observed in large convolutional networks [48].

Flatness of minima has been linked to good generalization [49]. The channels to infinity – being very flat in some directions, and progressively steeper in others – provide an interesting perspective on this: commonly used optimizers get stuck at the edge of stability [47], thus they appear to be flat local minima. Already at the beginning of the channel, the network implements (a finite difference approximation of) the gated linear unit, which can be beneficial for generalization. If many channels emerge during training, this could be an indication that the network architecture is suboptimal for the given task; a stylized example would be a teacher with gated linear units and a student with softplus neurons. Adding regularization has a similar effect as a finite step size: from a certain point the regularization outweighs the decrease in loss such that the dynamics get stuck inside the channel. Nonetheless, the network implements an approximation of the gated linear unit.

We envision our novel view on the loss landscape to be insightful for practical applications where expertise of the landscapes is crucial: model fusion [31, 33], adversarial robustness [50, 51], continual learning [52, 53] and federated learning [54, 34].

---

The code is available at `https://github.com/flavio-martinelli/channels-to-infinity`

## Acknowledgements

This work was supported by the Swiss National Science Foundation grant CRSII5 198612, 200020-207426 and 200021-236436. The authors would like to thank Antonio Orvieto for fruitful discussions on the edge of stability.

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

# A  Neuron duplication introduces lines of critical points

## A.1  Adding bias to neurons drastically reduces the number of stable plateau saddles

In the main text we highlighted specific attractive regions, plateau saddles, that can be found via neuron duplication. These attractive regions were first hypothesized theoretically by Fukumizu & Amari [1] for 1-hidden-layer, scalar-output MLPs. Figure 2 numerically explores the frequency with which plateau saddles are reached under the common Glorot normal initialization in the Fukumizu & Amari [1] setup without biases.

The stability of saddle lines is dependent on the positive (negative) definiteness of the $(d \times d)$-dimensional matrix: $B_j^r = \langle a_j \sigma''(\boldsymbol{w}_j \boldsymbol{x}) \boldsymbol{x} \boldsymbol{x}^T \frac{\partial}{\partial f(\boldsymbol{x})} \mathcal{L}_r(f(\boldsymbol{x}), f^*(\boldsymbol{x})) \rangle_{\mathcal{D}}$ ([1]; Theorem 3), where $\boldsymbol{w}_j, a_j$ are the input and output weights of neuron to duplicate, $\mathcal{L}_r$ is the loss of the original network, $f^*(\boldsymbol{x})$ is the target function, $f(\boldsymbol{x})$ is the network output, $\langle \cdot \rangle_{\mathcal{D}}$ denotes the expectation w.r.t. the distribution of the training data $p(\boldsymbol{x}) = \frac{1}{N} \sum_{i=1}^{N} \delta(\boldsymbol{x} - \boldsymbol{x}^{(i)})$, and $\boldsymbol{w}_j, \boldsymbol{x} \in \mathbb{R}^d$. It is difficult to determine in general whether the $B_j^r$ matrix is positive (negative) definite or not, and thus whether the saddle line is stable or unstable. Empirically, we notice that the simple addition of a bias term to the neurons drastically reduces the number of plateau saddles in the loss landscape, as shown in Figure A1 (in this case $B_j^r$ is $(d+1 \times d+1)$-dimensional).

The stability in multi-output and multi-layer settings has also been theoretically explored [12, 13, 7]. In addition to $B_j^r$ being positive (negative) definite, another condition must hold on a $(d_{l-1} \times d_{l+1})$-dimensional matrix $D$ being zero ([7]; Theorem 9), where $d_{l-1}$ and $d_{l+1}$ are the number of neurons in the previous and next layer respectively. In particular, Petzka & Sminchisescu [7] argue that the likelihood of all eigenvalues of $B_j^r$ being positive (negative) in conjunction with the likelihood of $D$ being zero in practical settings is extremely low. This conclusion is further supported by our numerical observations in Figure 4 where we find little to no exact duplicate neuron.

## A.2  Simulation details: Figure 2

We trained networks of 5 different architectures, with $r \in \{1, 2, 3, 4, 5\}$ neurons in a single hidden layer. To guarantee enough coverage of (presumably) all unique minima in the landscape, each architecture was simulated $50 \cdot 2^r$ times. Initializations were drawn from the Glorot normal distribution and we used the mean-squared error loss $\mathcal{L}(\boldsymbol{\theta}) = \langle [\sum_{i=1}^{r} a_i \sigma(\boldsymbol{w}_i \boldsymbol{x} + b_i) + c - f^*(\boldsymbol{x})]^2 \rangle_{\mathcal{D}}$, where $\boldsymbol{\theta} = (\boldsymbol{w}, \boldsymbol{a}, \boldsymbol{b}, c)$ and $\sigma(x) = \text{sigmoid}(4x) + \text{softplus}(x)$ is an asymmetric activation function introduced in [6, 23]. The target function $f^*(\boldsymbol{x})$ is a modified version of the 2D Rosenbrock function:

$$\tilde{f}^*(x_1, x_2) = \log_{10} \left[ (a - x_1)^2 + b(x_2 - x_1^2 + c)^2 + d \right]$$
$$f^*(x_1, x_2) = \text{zscore}_{\mathcal{D}}[\tilde{f}^*(x_1, x_2)] \tag{7}$$

where $a = 1$, $b = 3$, $c = 1$, $d = 0.1$ and $\text{zscore}_{\mathcal{D}}[f(\boldsymbol{x})] = \frac{f(\boldsymbol{x}) - \langle f(\boldsymbol{x}) \rangle_{\mathcal{D}}}{\sqrt{\langle [f(\boldsymbol{x}) - \langle f(\boldsymbol{x}) \rangle_{\mathcal{D}}]^2 \rangle_{\mathcal{D}}}}$. The modified Rosenbrock function was chosen due to its complicated, non-symmetric profile, leading a rich variety of solutions found by the networks.

Each simulation was performed on a single AMD EPYC 9454 48-Core Processor CPU core, using ODE solvers to solve the gradient flow equation: $\dot{\boldsymbol{\theta}} = -\nabla_{\boldsymbol{\theta}}(\mathcal{L}(\boldsymbol{\theta}) + R(\boldsymbol{\theta}))$ with the Julia package MLPGradientFlow.jl [44]. $R(\boldsymbol{\theta}) = \frac{1}{3}(||\boldsymbol{\theta}|| - \text{maxnorm})^3 \; if \; ||\boldsymbol{\theta}|| > \text{maxnorm}, else \; R(\boldsymbol{\theta}) = 0$, is a regularizer active only when a maxnorm threshold is reached. This allows us to verify convergence of our simulations and halt them when the norm exceeds too high values. The dataset consisted of $N$ samples drawn once for all seeds, with input distributed on a regular 2D grid with $x_1, x_2 \in [-\sqrt{3}, +\sqrt{3}]$. Training was performed full-batch, meaning that the only source of randomness is the initialization seed. Both input and output data have mean zero and standard deviation one. Hyperparameters of the simulations are provided in Table 1, where: `patience` is the number of iterations to wait before stopping the ODE solver if no improvement in the loss is observed, `reltol` and `abstol` are the relative and absolute tolerances for the ODE solver, `maxnorm` is the maximum norm of the gradient flow trajectory; we consider trajectories that exceed this norm as infinite-norm solutions.

Evidence for the precision of convergence of our simulations is shown in Figure A3. We report extremely low gradient norms and non-negative minimum eigenvalues for all simulated networks.

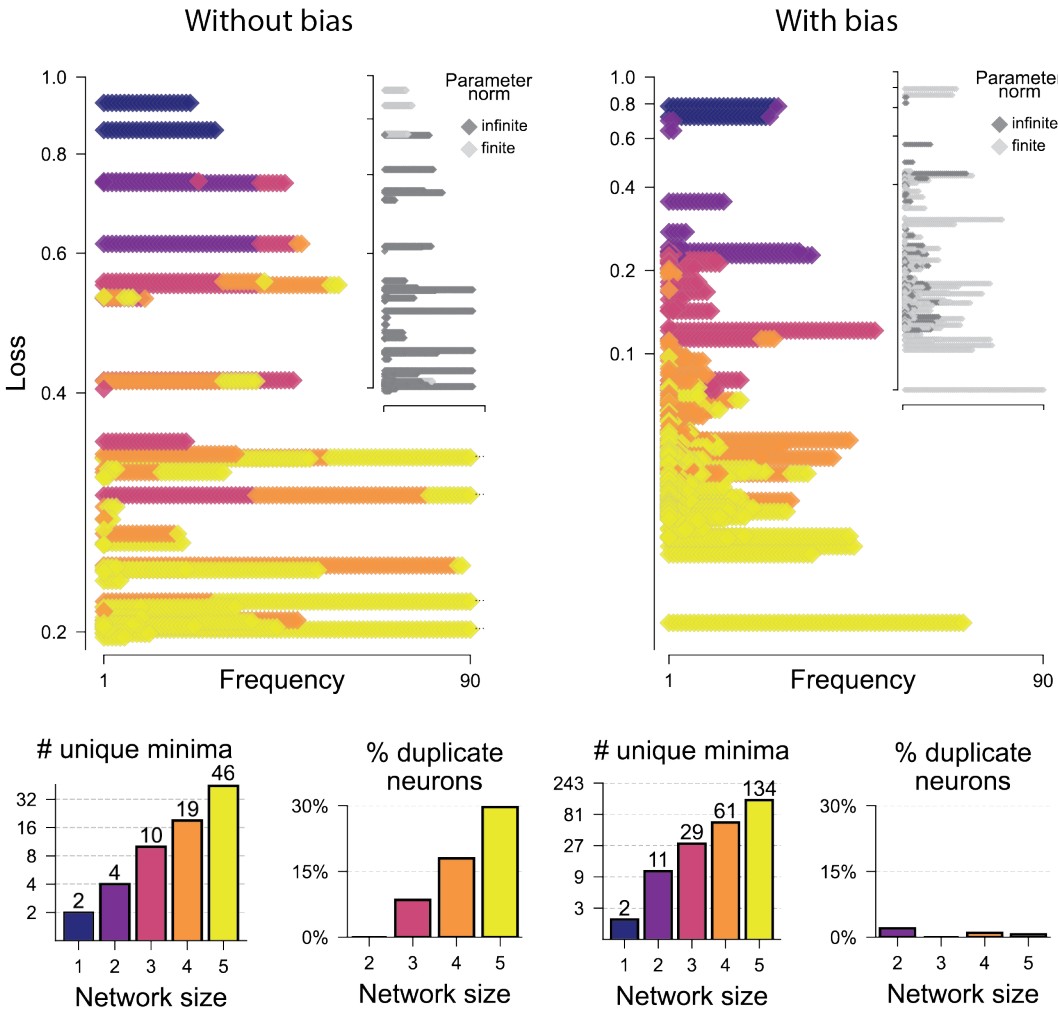

Figure A1: **Plateau saddles mostly disappear with the addition of a bias term:** side-by-side comparison between the experiment of Figure 2 and its version with neurons with bias. The amount of solutions sharing the same exact loss level across different network sizes drastically decreases when adding a bias term: horizontally overlapping solutions of different colors are barely present on the right panel. This is evidence that most of the saddle lines in the landscape do not contain stable regions, quantification of duplicate solutions (bottom) shows little to no presence of plateau saddles. In contrast, in both experiments, a substantial amount of solution trajectories are unbounded in parameter-norm (gray-scale insets).

The network used in Figure 2c,d was selected from the simulations of Figure 2a,b. Specifically, it is the $d = 2, r = 2$ network with finite parameter norm (see inset of left plot of Figure A1). Perturbations were performed after splitting the neuron with the negative output weight (blue arrow color in Figure A2a) at different $\gamma$ values: $\boldsymbol{\theta}^* \to \boldsymbol{\theta}^\gamma$. Figure 2d experiments show training convergence and trajectories of gradient flow after initializing at a point $\boldsymbol{\theta}^\gamma_\alpha$ near the saddle line. Specifically: $\boldsymbol{\theta}^\gamma_\alpha = \boldsymbol{\theta}^\gamma + \alpha \boldsymbol{e}_{\min}$, where $\boldsymbol{e}_{\min}$ is the eigenvector of the Hessian of $\boldsymbol{\theta}^\gamma$ with the minimum eigenvalue, and $\alpha = 10^{-7}$. The training procedure follows the same details as in subsection A.2, with the exception of using a slower ODE solver, `Heun` [55], for the early phase of the simulation. We chose `Heun` to obtain more trajectory steps near the saddle line. An interesting eigenvalue repulsion phenomenon is detailed in figure Figure A4.

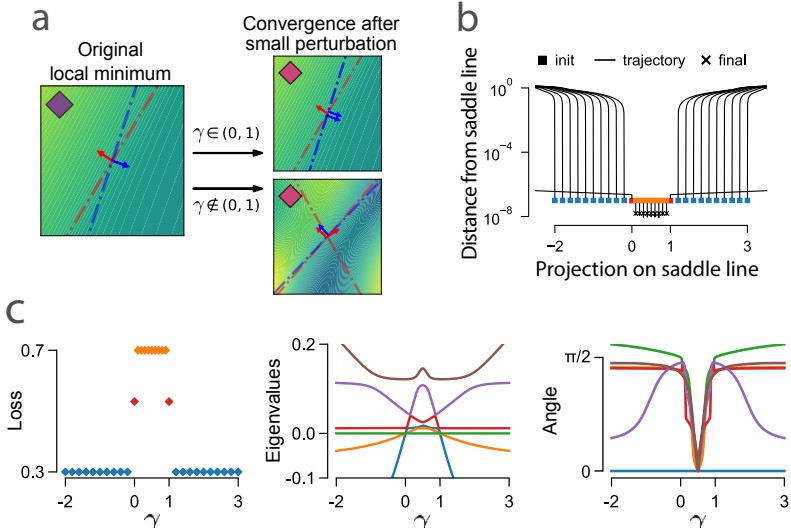

Figure A2: **Additional details for Figure 2.** (a) Example of neuron duplication for loss function shown in Figure 2b: small perturbations are stable only within the plateau-saddle region, $\gamma \in (0, 1)$. (b) Gradient-flow trajectories following a small perturbation from the saddle line in the direction of $\alpha e_{\min}$; identical to Figure 2d. (c) Losses reached after perturbations from different $\gamma$ values (left; colors as in panel b), eigenvalues of the Hessian as function of $\gamma$ (middle), and rotation angle of each eigenvector measured from the starting point $\gamma = 1/2$ (right).

Table 1: Hyperparameters of simulations in Figure 2.

| $N$ | maxtime | patience | reltol | abstol | maxnorm | ODE solver |
|-----|---------|----------|--------|--------|---------|------------|
| $10^4$ | 1h | $10^6$ | $10^{-3}$ | $10^{-6}$ | $10^3$ | KenCarp58 [56] |

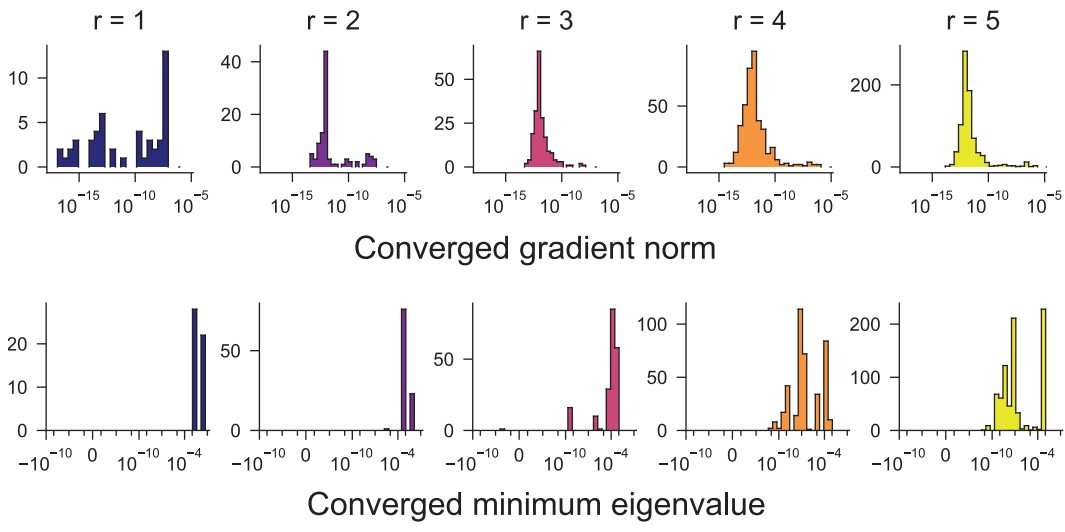

Figure A3: **Evidence of convergence to local minima or channels to infinity for simulations in Figure 2:** the $L_\infty$-norm $\max_i |\nabla_{\theta_i} \mathcal{L}(\theta)|$ is used to quantify the gradient norm.

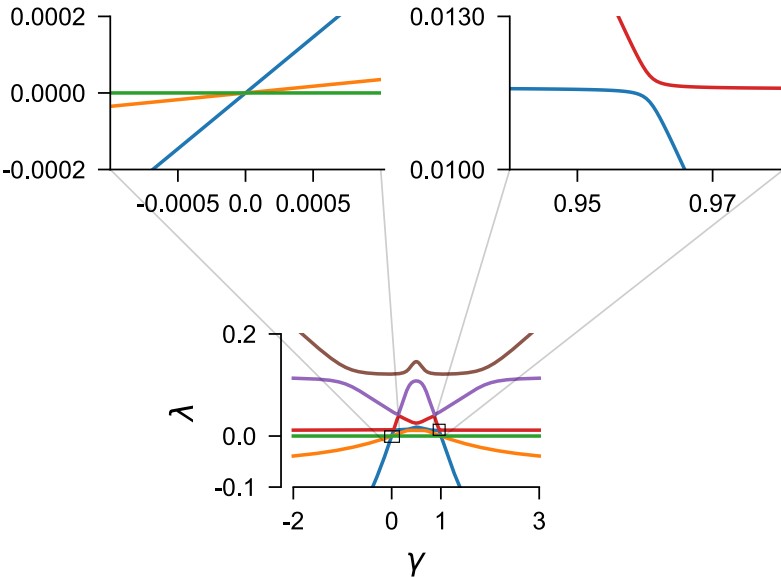

Figure A4: **Eigenvalue swapping if $\gamma \in \{0, 1\}$ and eigenvalue repulsion if $\gamma \notin \{0, 1\}$**: eigenvalues are colored based on continuity of their corresponding eigenvectors, as two eigenvalue approach each other and swap order, this is the only way to keep track of their identity. A particular characteristic of the saddle line is the extra degeneracy happening at $\gamma = 0$ and $\gamma = 1$. On top of the obvious zero-eigenvalue corresponding to the eigenvector in the direction of the saddle line (green line), another $d$ eigenvalues cross zero at $\gamma = 0$ and $\gamma = 1$. This $d$-dimensional space corresponds to the subspace spanned by the input weight vectors of the duplicate neuron that is being silenced (since $a_1 = \gamma a, a_2 = (1-\gamma)a$, either one of the two neurons is silenced at $\gamma = 0$ or $\gamma = 1$). For $\gamma \notin \{0, 1\}$, eigenvalues do not cross but repel each other. This phenomenon is a sign of non-degeneracy of the system for $\gamma \notin \{0, 1\}$; akin to eigenvalue repulsion phenomena in other fields [43].

# B Seemingly flat regions in the loss landscape as channels to infinity

## B.1 Simulation details: Figure 3

The perturbation procedure used in Figure 3 is similar to the one of Figure 2c,d, but with a different network. The network used in this example was selected from the simulations of Figure 4. We plotted the loss landscape of the network along the saddle line (see next paragraph for a discussion on how to find a saddle line from a given channel), and along the eigenvector $e_{\min}(\gamma)$ of the Hessian corresponding to the minimum eigenvalue. This eigenvector is computed at every value of $\gamma$ along the saddle line. Note that usually all eigenvectors rotate as $\gamma$ changes (Figure A2c), meaning that our loss surface is plotted along a rotating reference frame. The orange and green perturbations are performed by adding $\pm \alpha e_{\min}$, where $\alpha = 10^{-5}$. What is reported to be $\text{loss}_\infty$ in panel d is the loss of the network when the trajectory reaches the maxnorm. A zoomed-out view of the landscape, including negative values of $\gamma$ and the interval $\gamma \in [0, 1]$ is shown in Figure B5.

## B.2 Finding the closest saddle line from a given channel: an open challenge

Given the parameterization of a channel, identifying the parameterization of a saddle line parallel to said channel is not a trivial problem. As we have seen, usually saddle lines are non-attractive regions of the landscape, hence they cannot be found via simple optimization. Moreover, despite channels and saddle lines being asymptotically parallel subspaces, they are not necessarily close; all parameters can change along a trajectory that originates near a saddle line and ends in a channel (see early trajectories in Figure 3e and Figure B6). In order to find the parameterization of the saddle line from which a channel originated, we used the following heuristic: (i) From the converged solution in a channel, merge the pair of neurons with the closest cosine similarity in input weights. This is done by substituting the pair of neurons by a new neuron with the average input weight vector $w = (w_r + w_{r+1})/2$ and summed output weights $a = (a_r + a_{r+1})$. (ii) Train the resulting merged network with the same training procedure as in subsection A.2 for a few iterations. (iii) Verify that the parameterization reached, when duplicated, generates a saddle line that has some escape trajectories converging to the channel solution we started from. This procedure is not guaranteed to work, as in some cases step (ii) leads to other channel solutions, and/or step (iii) leads to different channels parallel to the one we started from. In conclusion, while generating channel solutions is straightforward (simply duplicate a neuron to create a saddle line then perturb), finding the parameterization of the saddle line that lead to a specific channel solution is not trivial. The example of Figure 3 is a successful case of this procedure. The network used was one of the networks obtained in the simulations of Figure 4.

## B.3 Simulation details: Figure 4

Five types of datasets were used in Figure 4: a multidimensional, modified version of the rosenbrock function and 4 gaussian processes (GP) with different kernel sizes. The modified d-dimensional rosenbrock function is defined as follows:

$$\tilde{f}^*(\boldsymbol{x}) = \log_{10}\left[\sum_{i=2}^{d}(a - x_{i-1})^2 + b(x_i - x_{i-1}^2 + c)^2 + d\right],$$

$$f^*(\boldsymbol{x}) = \text{zscore}_{\mathcal{D}}[\tilde{f}^*(\boldsymbol{x})], \tag{8}$$

where $a = 1$, $b = 3$, $c = 1$, $d = 0.1$. The Gaussian process datasets are generated using the `AbstractGPs.jl` package, using the Matern32 kernel:

$$k_s(\boldsymbol{x}, \boldsymbol{x}') = (1 + \sqrt{3}s\, d(\boldsymbol{x}, \boldsymbol{x}')) \exp(-\sqrt{3}s\, d(\boldsymbol{x}, \boldsymbol{x}')), \tag{9}$$

with $d(\cdot, \cdot)$ the Euclidean distance, and $s \in \{0.1, 0.5, 2, 10\}$ a scaling factor. Some examples of 2D GP datasets are shown in Figure B8. Given that channel solutions implement derivatives of the activation function, we were wondering whether their probability of convergence is related to the non-smoothness of the target function. Indeed this seems to be the case Figure 4b. All datasets were fitted with various architectures of MLPs with 1 hidden layer and $r \in \{2, 4, 8, 16\}$ neurons, for

---

In our exploration we found there exist multiple saddle lines parallel to a given channel. But not all of these, after small perturbations, lead the dynamics back to the original channel.

different input dimensions $d \in \{2, 4, 8, 16\}$. In particular, every GP dataset was re-drawn for every network, but kept the same across different initializations of the parameter vectors. Both the softplus and erf activation functions were used for these simulations. We also tested deeper network on the rosenbrock dataset, with 2 and 3 hidden layers with 4 neurons in each hidden layer and softplus activation function. For each configuration, we ran 50 fits with different random initializations of the parameter vectors. In total, we trained 4420 networks. The training followed the same procedure as in subsection A.2 and Table 1, converged gradient norms can be seen in Figure B7.

Throughout the analysis, we considered solutions as channels if they satisfy the following conditions:

1. a parameter norm larger than $10^3$ (see subsection A.2 for the definition of parameter norm),
2. a cosine distance between input weights $1 - \cos(\boldsymbol{w}_i \cdot \boldsymbol{w}_j / \|\boldsymbol{w}_i\| / \|\boldsymbol{w}_i\|)$ of less than $10^{-3}$,
3. and the fraction of parameter updates inside the channels subspace is bigger than $90\%$.

See Figure B9 for a version of panel a in Figure 4 computed for all datasets. The fraction of update inside the channels subspace in Figure 4b was computed as $\|\Delta\boldsymbol{\theta}_{\text{projected}}\|/\|\Delta\boldsymbol{\theta}\|$ for each network that we deemed to have reached a channel based on criteria 1 and 2. Here, $\Delta\boldsymbol{\theta}$ refers to the update of all parameters and $\Delta\boldsymbol{\theta}_{\text{projected}}$ refers to the update of parameters inside the subspace spanned by the channel. This subspace is spanned by the output weight dimensions of the neurons participating in the channel (the corresponding projection operator is $\mathbb{I}_{i \in \text{channel indices}}$) and the input weight vectors of the neurons participating in the channel (the corresponding projection operator is $\boldsymbol{W}_c(\boldsymbol{W}_c^\top \boldsymbol{W}_c)^{-1}\boldsymbol{W}_c^\top$ where $\boldsymbol{W}_c$ contains the channel input weight vectors). The analysis of Figure 4 was performed only by detecting channels with two participating neurons, but we noticed that some channel solutions contained triplets or even higher numbers of weights that were close to each other, leading to multi-dimensional channels. It also often happens that multiple pairs of weights are close to each other, leading to multiple one-dimensional channels. To detect multi-dimensional channels we fixed a set threshold of 0.01 for the cosine distance between input weights and then created groups of neuron indices in which each element satisfies the distance threshold with at least one element in the group (Figure B11). We ran this procedure on all networks that exceeded the maximum parameter norm. The size of these groups is reported in Figure 4g for all datasets. Finally, to detect channels between layers of deep MLPs in Figure 4f, we used the criteria 1 and 2 described at the beginning of this paragraph.

## B.4 Relu activation

Due to the non-differentiability of the relu activation function at 0, the associated landscape can contain cusps at which gradient flow dynamics get stuck. However, these cusps disappear in the infinite data limit (see subsection C.3); indeed, in this case there are examples where channels exist in relu networks (see Figure B13).

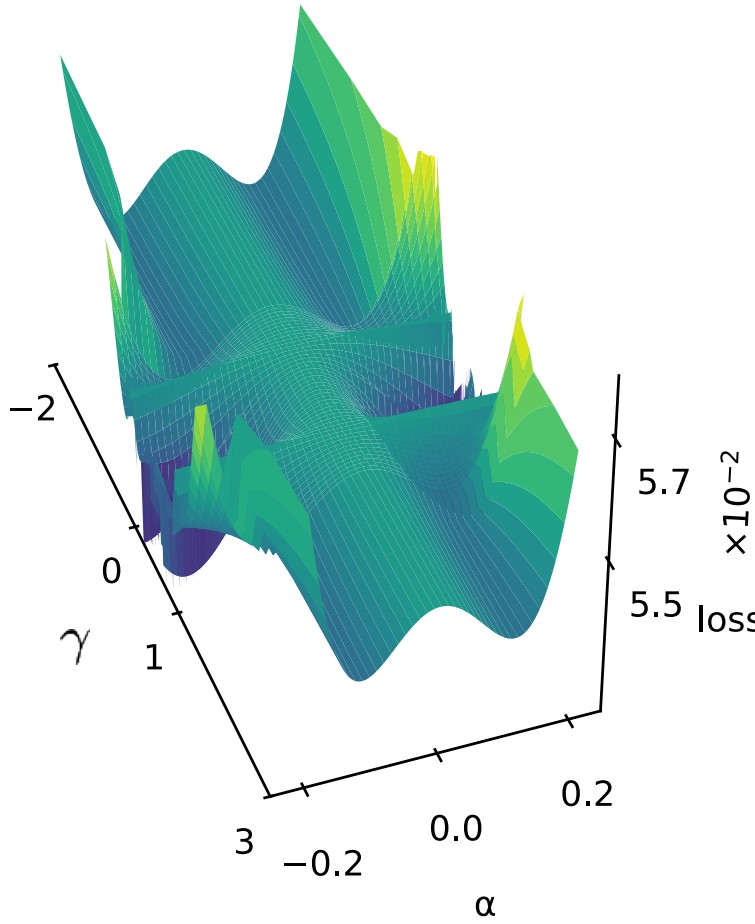

Figure B5: **Zoomed-out view of channel landscapes:** loss landscape of network of Figure 3. Many other structures appear in the interval $\gamma \in [0, 1]$ that are not explored in this paper. Note that the landscape is symmetric around the point $\gamma = 1/2$.

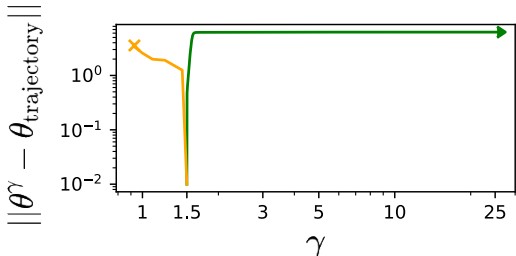

Figure B6: **Channels are parallel to saddle lines but not *near* them:** L2 distance of trajectories of Figure 3 to the saddle line. While the 2D picture of the landscape may give the illusion that channels a close to saddle lines, in reality there are many other dimensions that move substantially during the escape from the saddle.

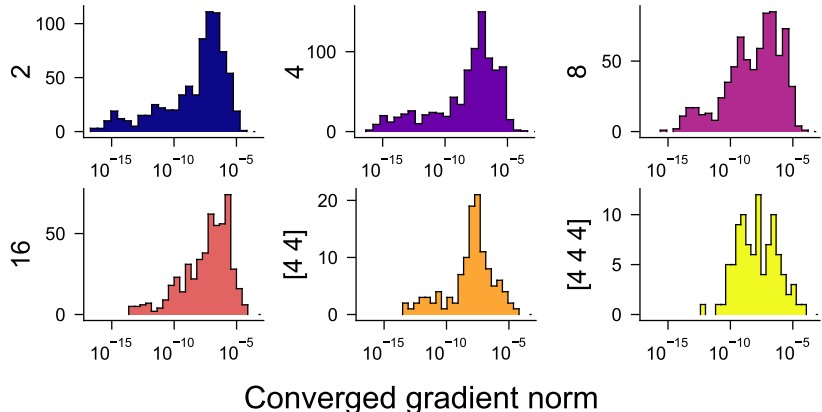

Figure B7: **Evidence of convergence to local minima or channels to infinity for simulations in Figure 4:** the $L_\infty$-norm $\max_i |\nabla_{\theta_i} \mathcal{L}(\boldsymbol{\theta})|$ is used to quantify the gradient norm.

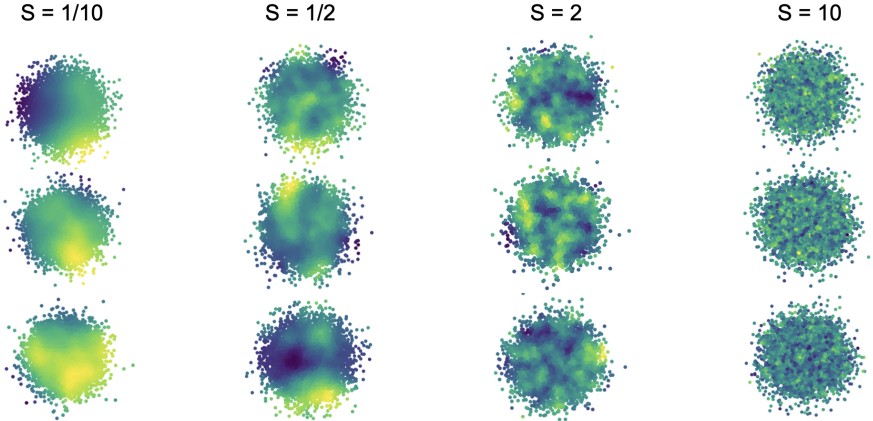

Figure B8: **Examples of 2D GP datasets:** 2D GP datasets used in Figure 4. The kernel size is controlled by the scaling factor $s$, the higher $s$, the bumpier is the dataset.

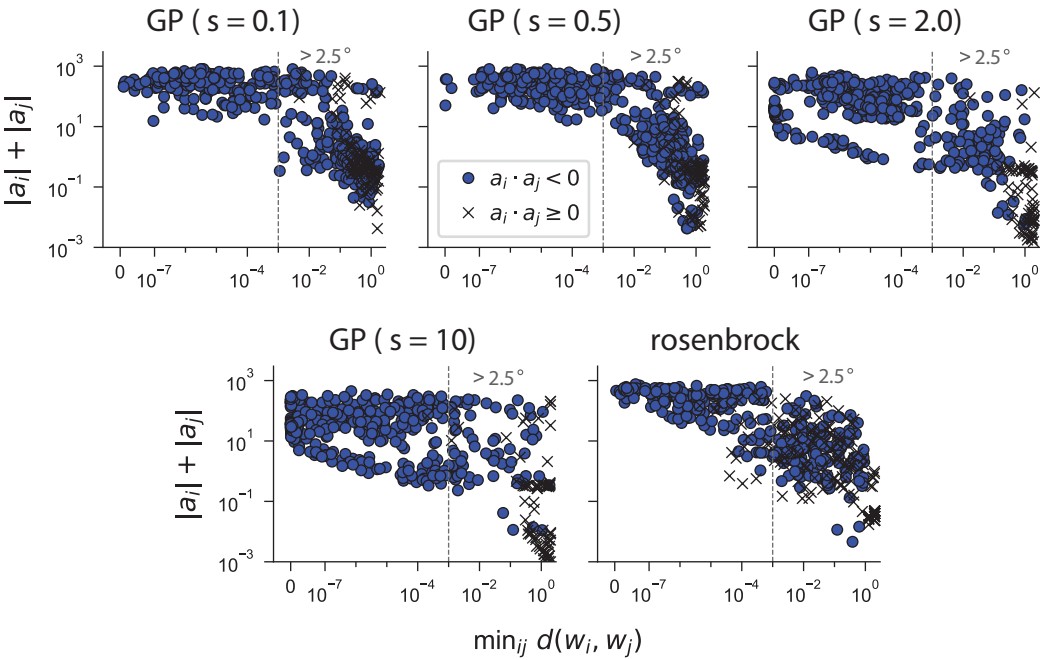

Figure B9: **Channel features in all datasets.**

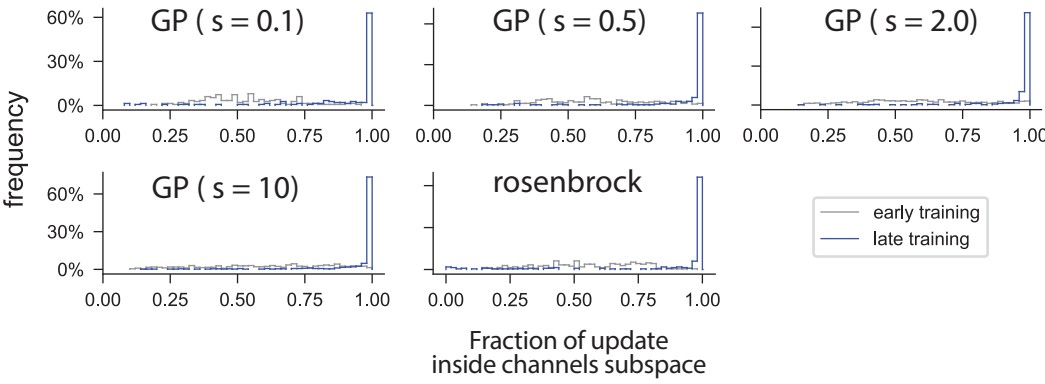

Figure B10: **Fraction of updates inside channels subspace (as in Figure 4b) across all datasets.**

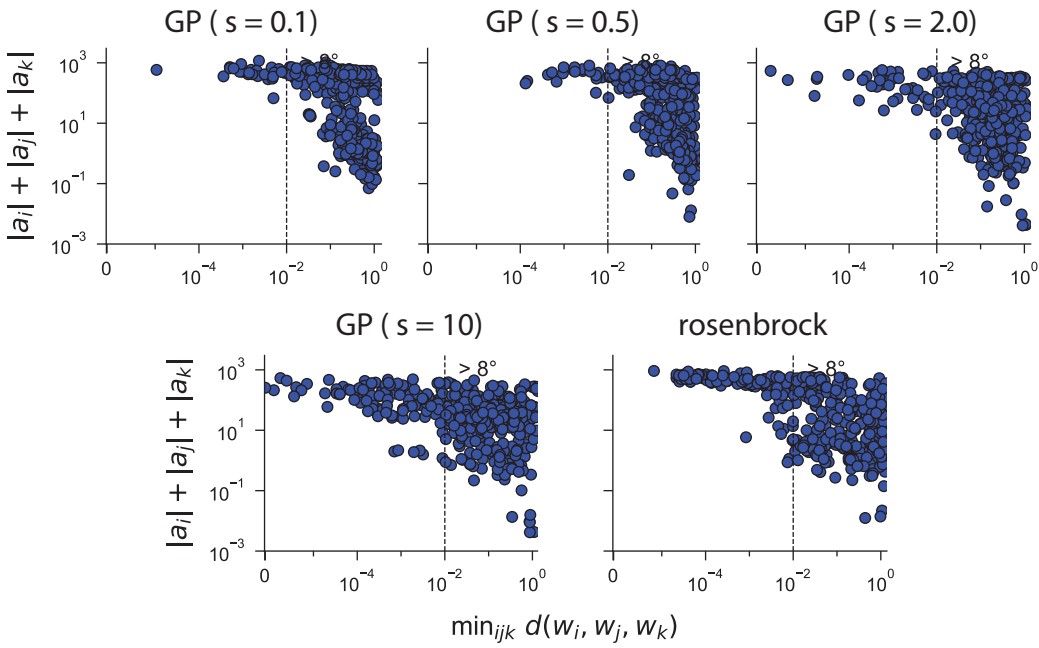

Figure B11: **2-dimensional channel features in all datasets, stemming from triplets of similar input weight vectors.**

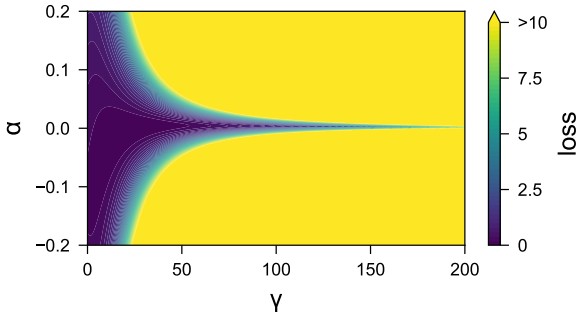

Figure B12: **View of a sharpening direction of the channel:** slice of the landscape for channel 2 of Figure 6 along $\Gamma$ and $\alpha e_{max}$, where $e_{max}$ is the eigenvector corresponding to the maximum eigenvalue along the channel $\lambda_{max}(\gamma)$.

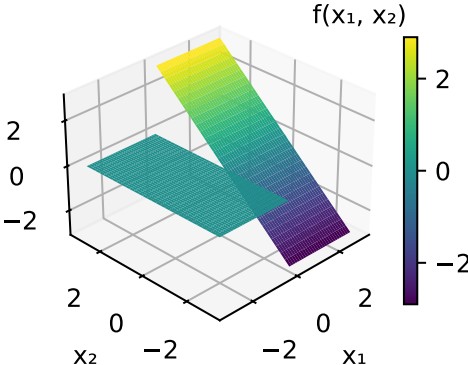

Figure B13: **Example of function learnable by a 2-2-1 relu MLP that converges in a channel**. The shown target function can be implemented by a relu MLP with 2 input, 2 hidden and 1 output neurons where $f(\boldsymbol{x}) = c\sigma(\boldsymbol{w} \cdot \boldsymbol{x}) + (\boldsymbol{v} \cdot \boldsymbol{x})\sigma'(\boldsymbol{w} \cdot \boldsymbol{x})$, with $c = 0$, $\boldsymbol{w} = [1, 0]$, $\boldsymbol{v} = [0, 1]$, $\sigma =$relu. In order to fit this function approximately with a 2-2-1 relu MLP we employed the infinite-data formulation of subsection C.3 with teacher weights $\boldsymbol{w}_1 = (1, 0)$, $\boldsymbol{w}_2 = (1, \epsilon)$ and $a_1 = 1/\epsilon, a_2 = -1/\epsilon$, with $\epsilon = 10^{-5}$.

## C Channels to infinity converge to gated linear units

### C.1 Jump procedure along channels to infinity

To move quickly along the channel, we iterate the following three steps multiple times:

1. Use the reparametrization below Equation 3 to obtain $c^{(t)}$, $a^{(t)}$, $\boldsymbol{w}^{(t)}$, $\boldsymbol{\Delta}^{(t)}$ and $\epsilon^{(t)}$ for given values of $a_i^{(t)}$, $a_j^{(t)}$, $\boldsymbol{w}_i^{(t)}$, $\boldsymbol{w}_j^{(t)}$ in step $t$ of the ODE solver.

2. Move approximately in the direction of the channel by lowering $\epsilon^{(t+1)} = \epsilon^{(t)}/2$, while keeping $c^{(t+1)} = c^{(t)}$, $a^{(t+1)} = a^{(t)}$, $\boldsymbol{w}^{t+1} = \boldsymbol{w}^{(t)}$, $\boldsymbol{\Delta}^{(t+1)} = \boldsymbol{\Delta}^{(t)}$. This point may not be at the "bottom of the channel", because the other parameters also move slightly when lowering $\epsilon$.

3. Compute the corresponding parameters $a_i^{(t+1)}$, $a_j^{(t+1)}$, $\boldsymbol{w}_i^{(t+1)}$, $\boldsymbol{w}_j^{(t+1)}$ and continue the ODE solver from this point to move again closer to the "bottom of the channel".

### C.2 Expansion of the loss in $\epsilon$

We start with the reparameterization in main text Equation 3,

$$a_i\sigma(\boldsymbol{w}_i \cdot \boldsymbol{x}) + a_j\sigma(\boldsymbol{w}_j \cdot \boldsymbol{x}) = \frac{c}{2}\left(\sigma\big((\boldsymbol{w} + \epsilon\boldsymbol{\Delta}) \cdot \boldsymbol{x}\big) + \sigma\big((\boldsymbol{w} - \epsilon\boldsymbol{\Delta}) \cdot \boldsymbol{x}\big)\right)$$
$$+ \frac{a}{2\epsilon}\left(\sigma\big((\boldsymbol{w} + \epsilon\boldsymbol{\Delta}) \cdot \boldsymbol{x}\big) - \sigma\big((\boldsymbol{w} - \epsilon\boldsymbol{\Delta}) \cdot \boldsymbol{x}\big)\right) \tag{10}$$

with $\boldsymbol{w} = (\boldsymbol{w}_i + \boldsymbol{w}_j)/2$, $\epsilon = \|\boldsymbol{w}_i - \boldsymbol{w}_j\|/2$, $\boldsymbol{\Delta} = \frac{\boldsymbol{w}_i - \boldsymbol{w}_j}{\|\boldsymbol{w}_i - \boldsymbol{w}_j\|}$, $a = \epsilon(a_i - a_j)$, and $c = a_i + a_j$. Performing a Taylor expansion in $\epsilon$ leads to

$$a_i\sigma(\boldsymbol{w}_i \cdot \boldsymbol{x}) + a_j\sigma(\boldsymbol{w}_j \cdot \boldsymbol{x}) = c\sigma(\boldsymbol{w} \cdot \boldsymbol{x}) + a(\boldsymbol{\Delta} \cdot \boldsymbol{x})\sigma'(\boldsymbol{w} \cdot \boldsymbol{x})$$
$$+ \epsilon^2\left(\frac{c}{2}(\boldsymbol{\Delta} \cdot \boldsymbol{x})^2\sigma''(\boldsymbol{w} \cdot \boldsymbol{x}) + \frac{a}{6}(\boldsymbol{\Delta} \cdot \boldsymbol{x})^3\sigma'''(\boldsymbol{w} \cdot \boldsymbol{x})\right) + \mathcal{O}(\epsilon^4). \tag{11}$$

Importantly, all contributions of $\mathcal{O}(\epsilon)$ cancel such that the leading corrections are $\mathcal{O}(\epsilon^2)$.

This implies that the network output is

$$f(\boldsymbol{x}; \boldsymbol{\theta}_0, \epsilon) = f_0(\boldsymbol{x}; \boldsymbol{\theta}_0) + \epsilon^2 f_2(\boldsymbol{x}; \boldsymbol{\theta}_0) + \mathcal{O}(\epsilon^4) \tag{12}$$

where $\boldsymbol{\theta}_0$ are the parameters for $\epsilon \to 0$, $f_0(\boldsymbol{x}; \boldsymbol{\theta}_0) = \lim_{\epsilon \to 0} f(\boldsymbol{x}; \boldsymbol{\theta}, \epsilon)$ is the output in the limit, and $f_2(\boldsymbol{x}; \boldsymbol{\theta}_0) = \lim_{\epsilon \to 0} \epsilon^{-2}[f(\boldsymbol{x}; \boldsymbol{\theta}_0, \epsilon) - f_0(\boldsymbol{x}; \boldsymbol{\theta}_0)]$ denotes the leading order correction. For a single-hidden-layer perceptron $f_2(\boldsymbol{x}; \boldsymbol{\theta}_0) = \frac{c}{2}(\boldsymbol{\Delta} \cdot \boldsymbol{x})^2\sigma''(\boldsymbol{w} \cdot \boldsymbol{x}) + \frac{a}{6}(\boldsymbol{\Delta} \cdot \boldsymbol{x})^3\sigma'''(\boldsymbol{w} \cdot \boldsymbol{x})$. Inserting (12) into the loss $\mathcal{L}_r(\boldsymbol{\theta}; \mathcal{D}) = \frac{1}{N}\sum_{i=1}^N \ell[f(\boldsymbol{x}^{(i)}; \boldsymbol{\theta}_0, \epsilon), y^{(i)}]$ and expanding in $\epsilon$ shows

$$\mathcal{L}_r(\boldsymbol{\theta}; \mathcal{D}) = \mathcal{L}_r(\boldsymbol{\theta}_0; \mathcal{D}) + \epsilon^2 h(\boldsymbol{\theta}_0; \mathcal{D}) + \mathcal{O}(\epsilon^4) \tag{13}$$

with $h(\boldsymbol{\theta}_0; \mathcal{D}) = \frac{1}{N}\sum_{i=1}^N f_2(\boldsymbol{x}^{(i)}; \boldsymbol{\theta}_0)\frac{\partial\ell}{\partial f}[f_0(\boldsymbol{x}^{(i)}; \boldsymbol{\theta}_0), y^{(i)}]$. We note that the expansion of the loss (13) is a direct consequence of the $\mathcal{O}(\epsilon^2)$ error of the central finite difference approximation (11).

### C.3 Infinite data (population) loss

To avoid finite data effects, we consider the population loss

$$\ell(\boldsymbol{\theta}) = \lim_{N \to \infty} \frac{1}{N}\sum_{i=1}^N \ell[f(\boldsymbol{x}^{(i)}; \boldsymbol{\theta}), f(\boldsymbol{x}^{(i)}; \boldsymbol{\theta}^*)] = \int \ell[f(\boldsymbol{x}; \boldsymbol{\theta}), f(\boldsymbol{x}; \boldsymbol{\theta}^*)]p(\boldsymbol{x})d\boldsymbol{x} \tag{14}$$

for a given data distribution $p(\boldsymbol{x}) = \lim_{N \to \infty} \frac{1}{N}\sum_{i=1}^N \delta(\boldsymbol{x} - \boldsymbol{x}^{(i)})$ and target function $f(\boldsymbol{x}; \boldsymbol{\theta}^*)$ implemented by a teacher network. To ease the notation, we denote expectations w.r.t. the data distribution $p(\boldsymbol{x})$ by angular brackets, $\langle f(\boldsymbol{x}) \rangle = \int f(\boldsymbol{x})p(\boldsymbol{x})d\boldsymbol{x}$; angular brackets with subscript $\mathcal{D}$ are reserved for finite data sets.

For a mean-squared error loss and a single-hidden-layer perceptron teacher and student networks, the population loss is

$$\ell(\boldsymbol{\theta}) = \frac{1}{2}\Big\langle \Big[\sum_{j=1}^{r^*} a_j^* \sigma(\boldsymbol{w}_j^* \cdot \boldsymbol{x} + b_j^*) - \sum_{j=1}^{r} a_j \sigma(\boldsymbol{w}_j \cdot \boldsymbol{x} + b_j)\Big]^2 \Big\rangle$$

$$= \frac{1}{2}\langle f(\boldsymbol{x};\boldsymbol{\theta}^*)^2 \rangle - \sum_{j=1}^{r^*}\sum_{k=1}^{r} a_j^* \langle \sigma(\boldsymbol{w}_j^* \cdot \boldsymbol{x} + b_j^*)\sigma(\boldsymbol{w}_k \cdot \boldsymbol{x} + b_k)\rangle a_k$$

$$+ \frac{1}{2}\sum_{j=1}^{r}\sum_{k=1}^{r} a_j \langle \sigma(\boldsymbol{w}_j \cdot \boldsymbol{x} + b_j)\sigma(\boldsymbol{w}_k \cdot \boldsymbol{x} + b_k)\rangle a_k \tag{15}$$

where we made the bias $b_j$ explicit, i.e., $\boldsymbol{x} \in \mathbb{R}^d$ for this subsection, and assumed matching activation function of teacher and student.

For a Gaussian data distributions $p(\boldsymbol{x}) = \prod_{i=1}^{d}\mathcal{N}(x_i \,|\, 0, 1)$ the preactivations $z_j = \boldsymbol{w}_j \cdot \boldsymbol{x}$ are Gaussian with zero mean and covariance $\langle z_j z_k \rangle = \boldsymbol{w}_j \cdot \boldsymbol{w}_k$. Thus, the expectations are a function of the biases and input weight overlaps,

$$\langle \sigma(\boldsymbol{w}_j \cdot \boldsymbol{x} + b_j)\sigma(\boldsymbol{w}_k \cdot \boldsymbol{x} + b_k)\rangle = g(b_j, b_k, \boldsymbol{w}_j \cdot \boldsymbol{w}_j, \boldsymbol{w}_j \cdot \boldsymbol{w}_k, \boldsymbol{w}_k \cdot \boldsymbol{w}_k). \tag{16}$$

For $\sigma(z) = \mathrm{erf}(z/\sqrt{2})$ and $\sigma(z) = \max(0, z)$ the function $g$ can be computed analytically (see C.3.1 and C.3.2). Inserting Equation 16 into Equation 15 leads to

$$\ell(\boldsymbol{\theta}) = \frac{1}{2}\langle f(\boldsymbol{x};\boldsymbol{\theta}^*)^2 \rangle - \sum_{j=1}^{r^*}\sum_{k=1}^{r} a_j^* a_k g(b_j^*, b_k, \boldsymbol{w}_j^* \cdot \boldsymbol{w}_j^*, \boldsymbol{w}_j^* \cdot \boldsymbol{w}_k, \boldsymbol{w}_k \cdot \boldsymbol{w}_k)$$

$$+ \frac{1}{2}\sum_{j=1}^{r}\sum_{k=1}^{r} a_j a_k g(b_j, b_k, \boldsymbol{w}_j \cdot \boldsymbol{w}_j, \boldsymbol{w}_j \cdot \boldsymbol{w}_k, \boldsymbol{w}_k \cdot \boldsymbol{w}_k) \tag{17}$$

We investigate the properties of the landscape using gradient flow, $\dot{\boldsymbol{\theta}} = -\nabla_{\boldsymbol{\theta}}\ell(\boldsymbol{\theta})$, where $\ell(\boldsymbol{\theta})$ is given by Equation 17.

### C.3.1  Gaussian expectations with error function

We want to compute the expectation (16) with $\sigma(z) = \mathrm{erf}(z/\sqrt{2})$. However, the expectation is simpler to compute for the cumulative standard normal distribution $G(z) = \frac{1}{2}[1 + \mathrm{erf}(z/\sqrt{2})]$; afterwards we use $\sigma(z) = 2G(z) - 1$ to get the expectation for the error function. For the case without biases the expectation is known [26]; for the case with biases the result is to the best of our knowledge new.

Without loss of generality we write $z_i = \frac{\sigma_i}{\sqrt{2}}(\sqrt{1+\rho}x \pm \sqrt{1-\rho}y)$ where $\boldsymbol{w}_i \cdot \boldsymbol{w}_i = \sigma_i^2$, $\boldsymbol{w}_i \cdot \boldsymbol{w}_j = \sigma_i \sigma_j \rho$, and $x, y$ are i.i.d. standard normal. Thus, we want to compute

$$\langle G(\mu_1 + z_1)G(\mu_2 + z_2)\rangle = \int_{-\infty}^{\infty} dx\, G'(x) \int_{-\infty}^{\infty} dy\, G'(y) G\Big[\mu_1 + \frac{\sigma_1(\sqrt{1+\rho}x + \sqrt{1-\rho}y)}{\sqrt{2}}\Big]$$

$$\times G\Big[\mu_2 + \frac{\sigma_2(\sqrt{1+\rho}x - \sqrt{1-\rho}y)}{\sqrt{2}}\Big] \tag{18}$$

where $G'(z) = \frac{1}{\sqrt{2\pi}}e^{-\frac{1}{2}z^2}$ is the standard normal distribution. We use formula 20,010.3 from Owen's table [57], $\int_{-\infty}^{\infty} dx\, G'(x)G(a+bx)G(c+dx) = \mathrm{BvN}\big(\frac{a}{\sqrt{1+b^2}}, \frac{c}{\sqrt{1+d^2}}; \frac{bd}{\sqrt{1+b^2}\sqrt{1+d^2}}\big)$ where $\mathrm{BvN}(h, k; \rho) = \frac{1}{2\pi\sqrt{1-\rho^2}}\int_{-\infty}^{h} dx \int_{-\infty}^{k} dy\, e^{-\frac{x^2 - 2\rho xy + y^2}{2(1-\rho^2)}}$ is the bivariate standard normal CDF, to get

$$\langle G(\mu_1 + z_1)G(\mu_2 + z_2)\rangle = \int_{-\infty}^{\infty} dx\, G'(x)$$

$$\times \mathrm{BvN}\Big(\frac{\mu_1 + \frac{\sigma_1\sqrt{1+\rho}}{\sqrt{2}}x}{\sqrt{1 + \frac{\sigma_1^2(1-\rho)}{2}}}, \frac{\mu_2 + \frac{\sigma_2\sqrt{1+\rho}}{\sqrt{2}}x}{\sqrt{1 + \frac{\sigma_2^2(1-\rho)}{2}}}; -\frac{\frac{\sigma_1\sigma_2(1-\rho)}{2}}{\sqrt{1 + \frac{\sigma_1^2(1-\rho)}{2}}\sqrt{1 + \frac{\sigma_2^2(1-\rho)}{2}}}\Big). \tag{19}$$

Next, we use that $\int_{-\infty}^{\infty} dx\, G'(x)\, \mathrm{BvN}(a+bx, c+dx; \rho) = \langle\langle\Theta(a+bx-y)\Theta(c+dx-z)\rangle_{y,z}\rangle_x$ where $y, z$ are bivariate Gaussian with zero mean, unit variance, and correlation $\langle yz \rangle = \rho$ and $\Theta(z)$ is the Heaviside step function. The random variables $\tilde{y} = \frac{y-bx}{\sqrt{1+b^2}}, \tilde{z} = \frac{z-dx}{\sqrt{1+d^2}}$ are still bivariate Gaussian with zero mean, unit variance, and correlation $\tilde{\rho} = \frac{\rho+bd}{\sqrt{1+b^2}\sqrt{1+d^2}}$. Thus, changing variables leads to the identity $\langle\langle\Theta(a+bx-y)\Theta(c+dx-z)\rangle_{y,z}\rangle_x = \langle\Theta(a-\sqrt{1+b^2}\tilde{y})\Theta(c-\sqrt{1+d^2}\tilde{z})\rangle_{\tilde{y},\tilde{z}}$. Using $\Theta(cx) = \Theta(x)$ yields

$$\int_{-\infty}^{\infty} dx\, G'(x)\, \mathrm{BvN}(a+bx, c+dx; \rho) = \mathrm{BvN}\left(\frac{a}{\sqrt{1+b^2}}, \frac{c}{\sqrt{1+d^2}}; \frac{\rho+bd}{\sqrt{1+b^2}\sqrt{1+d^2}}\right). \quad (20)$$

Note that 20,010.3 from [57] is the special case of (20) for $\rho = 0$ since $\mathrm{BvN}(a+bx, c+dx; 0) = G(a+bx)G(c+dx)$. Applying (20) to (19) leads to

$$\langle G(\mu_1+z_1)G(\mu_2+z_2)\rangle = \mathrm{BvN}\left(\frac{\mu_1}{\sqrt{1+\sigma_1^2}}, \frac{\mu_2}{\sqrt{1+\sigma_2^2}}; \rho\frac{\sigma_1\sigma_2}{\sqrt{1+\sigma_1^2}\sqrt{1+\sigma_2^2}}\right). \quad (21)$$

We use the identity (3.2) from [57], $\mathrm{BvN}(\mu_1, \mu_2; \rho) = T(\mu_1, \mu_2/\mu_1) + T(\mu_2, \mu_1/\mu_2) - T(\mu_1, \frac{\mu_2-\rho\mu_1}{\mu_1\sqrt{1-\rho^2}}) - T(\mu_2, \frac{\mu_1-\rho\mu_2}{\mu_2\sqrt{1-\rho^2}}) + G(\mu_1)G(\mu_2)$, to express $\mathrm{BvN}(\mu_1, \mu_2; \rho)$ in terms of Owen's T function $T(h, a) = \int_0^a \frac{G'(x)G'(hx)}{1+x^2} dx$ for an efficient numerical implementation of $\mathrm{BvN}(\mu_1, \mu_2; \rho)$ based on Gauss-Legendre quadrature.

For $\sigma(z) = \mathrm{erf}(z/\sqrt{2})$ we use $2G(z) - 1 = \mathrm{erf}(z/\sqrt{2})$, leading to

$$g(\mu_1, \mu_2, \sigma_1^2, \sigma_1\sigma_2\rho, \sigma_2^2) = 4\,\mathrm{BvN}\left(\frac{\mu_1}{\sqrt{1+\sigma_1^2}}, \frac{\mu_2}{\sqrt{1+\sigma_2^2}}; \rho\frac{\sigma_1\sigma_2}{\sqrt{1+\sigma_1^2}\sqrt{1+\sigma_2^2}}\right)$$
$$- 2G\left(\frac{\mu_1}{\sqrt{1+\sigma_1^2}}\right) - 2G\left(\frac{\mu_2}{\sqrt{1+\sigma_2^2}}\right) + 1 \quad (22)$$

where we used 10,010.8 from [57], $\int_{-\infty}^{\infty} dx\, G'(x)G(a+bx) = G(\frac{a}{\sqrt{1+b^2}})$.

### C.3.2 Gaussian expectations with ReLU function

We want to compute the expectation (16) with $\sigma(z) = \max(0, z)$. An alternative result can be found in [58]; this (simpler) expression is to the best of our knowledge new.

First, we use to the homogeneity $\sigma(\alpha z) = \alpha\sigma(z)$ to consider the case with unit variance; the variance simply multiplies the final expectation and modifies the mean as $\mu_i \to \mu_i/\sigma_i$. Following subsubsection C.3.1 we write $z_i = \frac{1}{\sqrt{2}}(\sqrt{1+\rho}x \pm \sqrt{1-\rho}y)$. Thus, we want to compute

$$\langle\sigma(\mu_1+z_1)\sigma(\mu_2+z_2)\rangle =$$
$$\frac{1}{2\pi\sqrt{1-\rho^2}}\int_{-\mu_1}^{\infty} dz_1 \int_{-\mu_2}^{\infty} dz_2\, e^{-\frac{1}{2(1-\rho^2)}[z_1^2-2\rho z_1 z_2+z_2^2]}(\mu_1+z_1)(\mu_2+z_2). \quad (23)$$

First, we shift integration variables, $z_i + \mu_i \to z_i$, to simplify the domain of integration in Equation 23 and rearrange terms to obtain

$$\langle\sigma(\mu_1+z_1)\sigma(\mu_2+z_2)\rangle =$$
$$\frac{1}{2\pi\sqrt{1-\rho^2}}\int_0^{\infty} dz_1 e^{-\frac{1}{2}(z_1-\mu_1)^2} z_1 \int_0^{\infty} dz_2\, e^{-\frac{1}{2(1-\rho^2)}[(z_2-\mu_2)-\rho(z_1-\mu_1)]^2} z_2. \quad (24)$$

The $z_2$ integral can be written as $\int_0^{\infty} dx\, G'\left(-\frac{\mu_2+\rho(z_1-\mu_1)}{\sqrt{1-\rho^2}} + \frac{x}{\sqrt{1-\rho^2}}\right)x$ for which we can use formula (101) from [57]: $\int_0^{\infty} dx\, G'(a+bx)x = \frac{1}{b^2}[G'(a) + aG(a) - a\Theta(b)]$. Using that $b > 0$ and $G(a) - 1 = -G(-a)$ this leads to

$$\langle\sigma(\mu_1+z_1)\sigma(\mu_2+z_2)\rangle = \int_0^{\infty} dz_1 e^{-\frac{1}{2}(z_1-\mu_1)^2} z_1 \left[\frac{\sqrt{1-\rho^2}}{2\pi} e^{-\frac{1}{2(1-\rho^2)}(\mu_2+\rho(z_1-\mu_1))^2}\right.$$
$$\left. + \frac{\mu_2+\rho(z_1-\mu_1)}{\sqrt{2\pi}} G\left(\frac{\mu_2-\rho\mu_1}{\sqrt{1-\rho^2}} + \frac{\rho}{\sqrt{1-\rho^2}}z_1\right)\right]. \quad (25)$$

Rearranging terms we obtain

$$\langle\sigma(\mu_1+z_1)\sigma(\mu_2+z_2)\rangle = \sqrt{1-\rho^2}G'(\mu_2)\int_0^\infty dz_1\, G'\big(-\frac{\mu_1-\rho\mu_2}{\sqrt{1-\rho^2}}+\frac{z_1}{\sqrt{1-\rho^2}}\big)z_1$$

$$+\mu_1\mu_2\int_{-\infty}^{\mu_1}dz_1\, G'(z_1)G\big(\frac{\mu_2-\rho z_1}{\sqrt{1-\rho^2}}\big)$$

$$+(\mu_2+\rho\mu_1)\int_{-\mu_1}^\infty dz_1\, G'(z_1)G\big(\frac{\mu_2}{\sqrt{1-\rho^2}}+\frac{\rho}{\sqrt{1-\rho^2}}z_1\big)z_1$$

$$+\rho\int_{-\mu_1}^\infty dz_1 G'(z_1)G\big(\frac{\mu_2}{\sqrt{1-\rho^2}}+\frac{\rho}{\sqrt{1-\rho^2}}z_1\big)z_1^2. \tag{26}$$

For the first integral we can use again formula (101), $\int_0^\infty dx\, G'(a+bx)x = \frac{1}{b^2}[G'(a)-aG(-a)]$ for $b>0$, for the second we need (10,010.2), $\int_{-\infty}^h dx\, G'(x)G(\frac{k-\rho x}{\sqrt{1-\rho^2}}) = \text{BvN}(h,k;\rho)$, for the third (10,011.1), $\int_c^\infty dx\, G'(x)G(a+bx)x = \frac{b}{\sqrt{1+b^2}}G'(\frac{a}{\sqrt{1+b^2}})G(-\frac{ab+(1+b^2)c}{\sqrt{1+b^2}}) + G(a+bc)G'(c)$, and for the fourth (10,01n.2) with help from [59] to get rid of typos, $\int_c^\infty dx\, G'(x)G(a+bx)x^2 = \frac{1}{2}G(\frac{a}{\sqrt{1+b^2}}) - \frac{1}{2}G(c) + T(c,b+\frac{a}{c}) + T(\frac{a}{\sqrt{1+b^2}},b+\frac{c(1+b^2)}{a}) + \frac{1}{2}\Theta(-a) + \frac{b}{1+b^2}G'(\frac{a}{\sqrt{1+b^2}})[-\frac{ab}{\sqrt{1+b^2}}G(-\frac{ab+(1+b^2)c}{\sqrt{1+b^2}}) + G'(\frac{ab+(1+b^2)c}{\sqrt{1+b^2}})] + cG'(c)G(a+bc)$ for $c>0$. This leads to

$$\langle\sigma(\mu_1+z_1)\sigma(\mu_2+z_2)\rangle = \frac{\sqrt{1-\rho^2}}{2\pi}e^{-\frac{\mu_1^2-2\rho\mu_1\mu_2+\mu_2^2}{2(1-\rho^2)}} + \mu_1\mu_2\text{BvN}(\mu_1,\mu_2;\rho)$$

$$+\mu_1 G'(\mu_2)G(\frac{\mu_1-\rho\mu_2}{\sqrt{1-\rho^2}}) + \mu_2 G'(\mu_1)G(\frac{\mu_2-\rho\mu_1}{\sqrt{1-\rho^2}})$$

$$+\rho\Big[\frac{1}{2}G(\mu_1)+\frac{1}{2}G(\mu_2) - T(\mu_1,\frac{\frac{\mu_2}{\mu_1}-\rho}{\sqrt{1-\rho^2}}) - T(\mu_2,\frac{\frac{\mu_1}{\mu_2}-\rho}{\sqrt{1-\rho^2}}) - \frac{1}{2}\Theta(\mu_2)\Big] \tag{27}$$

for $\mu_1<0$. Using formula (3.1) from [57] for $\text{BvN}(\mu_1,\mu_2;\rho)$ the expression reduces to

$$\langle\sigma(\mu_1+z_1)\sigma(\mu_2+z_2)\rangle = \frac{\sqrt{1-\rho^2}}{2\pi}e^{-\frac{\mu_1^2-2\rho\mu_1\mu_2+\mu_2^2}{2(1-\rho^2)}} + (\mu_1\mu_2+\rho)\text{BvN}(\mu_1,\mu_2;\rho)$$

$$+\mu_1 G'(\mu_2)G(\frac{\mu_1-\rho\mu_2}{\sqrt{1-\rho^2}}) + \mu_2 G'(\mu_1)G(\frac{\mu_2-\rho\mu_1}{\sqrt{1-\rho^2}}) \tag{28}$$

which works for arbitrary sign of $\mu_1$.

### C.3.3 Minimum at infinity

Here, we derive the stability condition for the minimum at the end of a channel. To this end, we consider the simplified setting where the input is scalar, $x\in\mathbb{R}$, and the network consists of two neurons without biases, leaving four parameters: $a_1,a_2,w_1,w_2$. We work in the reparameterization from Equation 10 which is for scalar input $w=(w_1+w_2)/2$, $\epsilon=(w_2-w_2)/2$, $a=\epsilon(a_1-a_2)$, and $c=a_1+a_1$. Together with Equation 11 we get $f(x;\boldsymbol{\theta}_0,\epsilon)=f_0(x;\boldsymbol{\theta}_0)+\epsilon^2 f_2(x;\boldsymbol{\theta}_0)+\mathcal{O}(\epsilon^4)$ with

$$f_0(x;\boldsymbol{\theta}_0) = c\sigma(wx)+ax\sigma'(wx), \tag{29}$$

$$f_2(x;\boldsymbol{\theta}_0) = \frac{c}{2}x^2\sigma''(wx)+\frac{a}{6}x^3\sigma'''(wx), \tag{30}$$

and $\boldsymbol{\theta}_0=(a,c,w)$. Note that for scalar input $\Delta=1$ and $\epsilon\in\mathbb{R}$ instead of $\epsilon\geq 0$. Expanding the loss (15) in $\epsilon$ we obtain

$$\ell(\boldsymbol{\theta}_0,\epsilon) = \frac{1}{2}\langle[f_0(x;\boldsymbol{\theta}_0)-f(x;\boldsymbol{\theta}^*)]^2\rangle + \epsilon^2\langle f_2(x;\boldsymbol{\theta}_0)[f_0(x;\boldsymbol{\theta}_0)-f(x;\boldsymbol{\theta}^*)]\rangle + \mathcal{O}(\epsilon^4), \tag{31}$$

reflecting the structure of Equation 13.

To reduce the number of parameters from $\boldsymbol{\theta}_0=(a,c,w)$ to $w$ we assume a separation of time scales: the dynamics of the readout weights $a,c$ are instantaneous in relation to the dynamics of the input

weights $w, \epsilon$. While the separation of time scales changes the details of the trajectory, it leaves the critical points invariant. In particular, this means that a stable minimum at $\epsilon \to 0$ with fast readout weights implies a stable minimum at $\epsilon \to 0$ without separation of time scales.

Under the assumption of separate time scales, the readout weights $a, c$ minimize the loss (31) for any given set of input weights $w, \epsilon$. Since (31) is quadratic in $a, c$ the minimization is straightforward to perform. Furthermore, it is sufficient to consider the minimization of the $\mathcal{O}(1)$ contribution because the $\mathcal{O}(\epsilon^2)$ correction to the loss due to an $\mathcal{O}(\epsilon^2)$ correction to $a, c$ vanishes by construction of the minimum. Thus, we get $\ell(w, \epsilon) = \ell(w, 0) + \epsilon^2 h(w) + \mathcal{O}(\epsilon^4)$ where

$$\ell(w, 0) = \frac{1}{2} \langle [f_0(x; \boldsymbol{\theta}_0(w)) - f(x; \boldsymbol{\theta}^*)]^2 \rangle, \tag{32}$$

$$h(w) = \langle f_2(x; \boldsymbol{\theta}_0(w)) [f_0(x; \boldsymbol{\theta}_0(w)) - f(x; \boldsymbol{\theta}^*)] \rangle, \tag{33}$$

with $\boldsymbol{\theta}_0(w) = (a_0(w), c_0(w), w)$ and $a_0(w)$ and $c_0(w)$ determined by the linear system

$$a_0 \langle x^2 \sigma'(wx)^2 \rangle + c_0 \langle \sigma(wx) x \sigma'(wx) \rangle = \langle f(x; \boldsymbol{\theta}^*) x \sigma'(wx) \rangle, \tag{34}$$

$$a_0 \langle x \sigma'(wx) \sigma(wx) \rangle + c_0 \langle \sigma(wx)^2 \rangle = \langle f(x; \boldsymbol{\theta}^*) \sigma(wx) \rangle. \tag{35}$$

All expectations are w.r.t. a one dimensional standard normal distribution which can be solved efficiently using Gauss-Hermite quadrature such that $\ell(w, 0)$ and $h(w)$ can be determined numerically. Stability requires $h(w) > 0$ at a minimum of $\ell(w, 0)$.

## C.4  Simulation details: Figure 5

We picked arbitrarily one of the networks with $r = 8$ hidden neurons (softplus activation function) with biases (81 parameters in total), trained on the modified Rosenbrock function in 8D (see Equation 8), which had a small minimal cosine distance $d(\boldsymbol{w}_i, \boldsymbol{w}_j)$ and large $|a_i| + |a_j|$ (see top left in Figure 4A). Starting with these parameter values, we applied the jump procedure described in Jump procedure along channels to infinity to obtain the results in Figure 5A.

## C.5  Simulation details: Figure 6

For the 4-dimensional toy example in Figure 5B we rely on Equation 17 with the analytical expression of $g$ from Equation 22, and derivatives thereof, to solve the gradient dynamics and compute the loss in the population average (infinity data) setting. The plot of the loss function (middle of Figure 5B) is produced by fixing $\gamma$ and $\alpha$ (the projection onto the eigenvector with the smallest eigenvalue on the saddleline at $\gamma \approx 2$), and minimizing the loss in the orthogonal, remaining 2 dimensions. The trajectories are orthogonally projected onto the $\gamma - \alpha$-plane. The curves for finite data (gray) were produced with 4096 standard normally distributed input samples that were fixed throughout the simulation. The same dataset was used to run SGD (learning rate $\eta = 0.1$) and ADAM (default parameters $\eta = 0.001, \beta_1 = 0.9, \beta_2 = 0.999, \epsilon = 10^{-8}$) with batchsize 16. We ran SGD and ADAM for a fixed duration, which resulted in approximately 50'000 training epochs. To determine the stable region (green in Figure 5B right), we used Gauss-Hermite quadrature to compute the integrals in Equation 33 for different values of $w$.

## C.6  Second-order derivatives

Obtaining higher order derivatives from multiple neurons requires a hierarchy of divergent readout weights and convergent input weights: a second order derivative leads to an $\mathcal{O}(\epsilon^2)$ contribution in the finite difference which requires a readout direction which diverges $\sim \epsilon^{-2}$ to get an $\mathcal{O}(1)$ contribution to the output in the limit. To capture this hierarchical structure, we change notation in this subsection compared to the main text.

We consider a single-hidden-layer perceptron with $r$ hidden units

$$f(\boldsymbol{x}; \boldsymbol{\theta}) = \boldsymbol{a} \cdot \sigma(\boldsymbol{W} \cdot \boldsymbol{x}) \tag{36}$$

with $\boldsymbol{a} \in \mathbb{R}^r$, $\boldsymbol{x} \in \mathbb{R}^{d+1}$, $\boldsymbol{W} \in \mathbb{R}^{r \times (d+1)}$, and $\sigma$ is applied element-wise. As before the $r$ neurons in this network can be part of a bigger network with multiple layers.

The divergent readout weights need a set of orthogonal axes $\{\boldsymbol{u}_\rho\}_{\rho=0}^{r-1}$ along which they can diverge with different speeds without influencing each other. We perform a change to the orthonormal basis

given by $\{\boldsymbol{u}_\rho\}_{\rho=0}^{r-1}$. In this new bases the hierarchy of divergent readout weights and convergent input weights is

$$\epsilon^\rho \boldsymbol{\omega}_\rho = \boldsymbol{W} \cdot \boldsymbol{u}_\rho, \qquad \epsilon^{-\rho} \alpha_\rho = \boldsymbol{u}_\rho \cdot \boldsymbol{a}, \tag{37}$$

where $\|\boldsymbol{\omega}_\rho\| = \mathcal{O}(1)$ and $\alpha_\rho = \mathcal{O}(1)$. Inserting Equation 37 into Equation 36 leads to

$$f(\boldsymbol{x}; \boldsymbol{\theta}_0, \epsilon) = \sum_{\rho=0}^{r-1} \frac{1}{\epsilon^\rho} \alpha_\rho \boldsymbol{u}_\rho \cdot \sigma\Big( \sum_{\rho'=0}^{r-1} \epsilon^{\rho'} (\boldsymbol{\omega}_{\rho'} \cdot \boldsymbol{x}) \boldsymbol{u}_{\rho'} \Big) \tag{38}$$

where $\boldsymbol{\theta}_0 = \{\alpha_\rho, \boldsymbol{\omega}_\rho\}_{\rho=0}^{r-1}$. Eventually the $\rho$-th term in the series will give rise to the $\rho$-th derivative in the limit $\epsilon \to 0$.

### C.6.1 First derivative with two neurons

For the familiar result of a first derivative with a two neuron network we change basis to

$$\boldsymbol{u}_0 = \frac{1}{\sqrt{2}} \begin{pmatrix} 1 \\ 1 \end{pmatrix}, \qquad \boldsymbol{u}_1 = \frac{1}{\sqrt{2}} \begin{pmatrix} 1 \\ -1 \end{pmatrix}. \tag{39}$$

This leads with Equation 38 to

$$f(\boldsymbol{x}; \boldsymbol{\theta}_0, \epsilon) = \sum_{\rho=0}^{1} \frac{1}{\epsilon^\rho} \alpha_\rho \boldsymbol{u}_\rho \cdot \sigma[(\boldsymbol{\omega}_0 \cdot \boldsymbol{x})\boldsymbol{u}_0 + \epsilon(\boldsymbol{\omega}_1 \cdot \boldsymbol{x})\boldsymbol{u}_1]$$

$$= \sum_{\rho=0}^{1} \frac{1}{\epsilon^\rho} \alpha_\rho \sum_{i=0}^{1} u_{i\rho}[\sigma(u_{i0}\boldsymbol{\omega}_0 \cdot \boldsymbol{x}) + \epsilon u_{i1}(\boldsymbol{\omega}_1 \cdot \boldsymbol{x})\sigma'(u_{i0}\boldsymbol{\omega}_0 \cdot \boldsymbol{x}) + \mathcal{O}(\epsilon^2)]$$

where we wrote the inner product explicitly in the second line and expanded in $\epsilon$. Since $u_{i0} = 1/\sqrt{2}$ does not depend on $i$ the inner product in the second term reduces to the inner product $\boldsymbol{u}_\rho \cdot \boldsymbol{u}_1 = \delta_{\rho 1}$; in combination with $\sum_{i=0}^{1} u_{i\rho} = \sqrt{2}\delta_{\rho 0}$ we get

$$f(\boldsymbol{x}; \boldsymbol{\theta}_0, \epsilon) = \sqrt{2}\alpha_0 \sigma(\boldsymbol{\omega}_0 \cdot \boldsymbol{x}/\sqrt{2}) + \alpha_1(\boldsymbol{\omega}_1 \cdot \boldsymbol{x})\sigma'(\boldsymbol{\omega}_0 \cdot \boldsymbol{x}/\sqrt{2}) + \mathcal{O}(\epsilon^2). \tag{40}$$

Note that the error is indeed $\mathcal{O}(\epsilon^2)$ because $\sum_{i=0}^{1} u_{i1}^3 = 0$. Connecting this result with the notation in the main text, we see $c = \sqrt{2}\alpha_0$, $a = \alpha_1$, $\boldsymbol{w} = \boldsymbol{\omega}_0/\sqrt{2}$, and $\boldsymbol{\Delta} = \boldsymbol{\omega}_1$.

### C.6.2 Second Derivative with Three Neurons

For the second derivative with a three neuron network we change basis to

$$\boldsymbol{u}_0 = \frac{1}{\sqrt{3}} \begin{pmatrix} 1 \\ 1 \\ 1 \end{pmatrix}, \qquad \boldsymbol{u}_1 = \frac{1}{\sqrt{2}} \begin{pmatrix} 1 \\ 0 \\ -1 \end{pmatrix}, \qquad \boldsymbol{u}_2 = \frac{1}{\sqrt{6}} \begin{pmatrix} 1 \\ -2 \\ 1 \end{pmatrix}. \tag{41}$$

Now we have an $\mathcal{O}(\epsilon^2)$ contribution in the preactivation, $(\boldsymbol{\omega}_0 \cdot \boldsymbol{x})\boldsymbol{u}_0 + \epsilon(\boldsymbol{\omega}_1 \cdot \boldsymbol{x})\boldsymbol{u}_1 + \epsilon^2(\boldsymbol{\omega}_2 \cdot \boldsymbol{x})\boldsymbol{u}_2$, which leads to

$$f(\boldsymbol{x}) = \sum_{\rho=0}^{2} \frac{1}{\epsilon^\rho} \alpha_\rho \sum_{i=0}^{2} u_{i\rho}\Big[\sigma(u_{i0}\boldsymbol{\omega}_0 \cdot \boldsymbol{x}) + \epsilon u_{i1}(\boldsymbol{\omega}_1 \cdot \boldsymbol{x})\sigma'(u_{i0}\boldsymbol{\omega}_0 \cdot \boldsymbol{x})$$

$$+ \epsilon^2 u_{i2}(\boldsymbol{\omega}_2 \cdot \boldsymbol{x})\sigma'(u_{i0}\boldsymbol{\omega}_0 \cdot \boldsymbol{x}) + \frac{1}{2}\epsilon^2 u_{i1}^2(\boldsymbol{\omega}_1 \cdot \boldsymbol{x})^2\sigma''(u_{i0}\boldsymbol{\omega}_0 \cdot \boldsymbol{x}) + \mathcal{O}(\epsilon^3)\Big]$$

Using again ortonormality $\boldsymbol{u}_\rho \cdot \boldsymbol{u}_1 = \delta_{\rho 1}$ in combination with $u_{i0} = 1/\sqrt{3}$, $\sum_{i=0}^{2} u_{i\rho} = \sqrt{3}\delta_{\rho 0}$, and $\sum_{i=0}^{2} u_{i2}u_{i1}^2 = 1/6$ we get in the limit $\epsilon \to 0$

$$f(\boldsymbol{x}) = \sqrt{3}\alpha_0 \sigma(\boldsymbol{\omega}_0 \cdot \boldsymbol{x}/\sqrt{3}) + [\alpha_1(\boldsymbol{\omega}_1 \cdot \boldsymbol{x}) + \alpha_2(\boldsymbol{\omega}_2 \cdot \boldsymbol{x})]\sigma'(\boldsymbol{\omega}_0 \cdot \boldsymbol{x}/\sqrt{3})$$

$$+ \frac{1}{12}\alpha_2(\boldsymbol{\omega}_1 \cdot \boldsymbol{x})^2\sigma''(\boldsymbol{\omega}_0 \cdot \boldsymbol{x}/\sqrt{3}). \tag{42}$$

Note that it is necessary to have the $\boldsymbol{\omega}_2 \cdot \boldsymbol{x}$ contribution to be $\mathcal{O}(\epsilon^2)$, otherwise the $\alpha_2(\boldsymbol{\omega}_2 \cdot \boldsymbol{x})$ term would diverge with $1/\epsilon$.

## Appendix References

[55] Endre Süli and David F Mayers. *An introduction to numerical analysis*. Cambridge university press, 2003.

[56] Christopher A Kennedy and Mark H Carpenter. Higher-order additive runge–kutta schemes for ordinary differential equations. *Applied numerical mathematics*, 136:183–205, 2019.

[57] Donald B. Owen. A table of normal integrals. *Commun. Stat. Simul. Comput.*, 9(4):389–419, 1980.

[58] Kirsten Fischer, Alexandre René, Christian Keup, Moritz Layer, David Dahmen, and Moritz Helias. Decomposing neural networks as mappings of correlation functions. *Phys. Rev. Res.*, 4:043143, Nov 2022.

[59] Donald B. Owen. A special case of a bivariate non-central t-distribution. *Biometrika*, 52(3/4):437–446, 1965.

