# OpenReview forum: "Flat Channels to Infinity in Neural Loss Landscapes"
_NeurIPS.cc/2025/Conference — NeurIPS 2025 poster_

### Official Review · Reviewer_eBLi · 2025-06-22

**Clarity:** 3
**Significance:** 4
**Originality:** 3
**Rating:** 5
**Confidence:** 3

**Summary:**

Training neural networks occurs on a surface called the *Loss Landscape*.
This paper zooms on regions related to minima of subnetworks and uncovers a novel structure called *Channel to infinity*.
Duplicating a neuron in a subnetwork yields a symmetry with degree(s) of freedom on the output of the duplicated neurons.
At a subnetwork minima, this process generates a saddle line which in turn gives rise to the channels on both sides of unstable regions of the saddle line.
Channels, having a slowly decreasing loss, push the parameter under gradient to infinity by making the output weights of two neurons infinite while equalizing their input weight vectors.
Moreover, at infinity the two entangled neurons simplify to a GLU unit.

Alongside a description of the phenonemon, the paper conducts various empirical analyses on the stability of the saddle line, the natural reachability of the channels, the different properties a pair of channels can exhibit, their flatness/sharpness and descriptive statistics touching on frequency or location in an architecture of entangled neurons.

**Questions:**

- You study plateau-saddles and networks that have neurons with equal input weight vectors. Do you think this relation could be generalized, e.g. 3 neurons which would perform computations doable by 2 (but where no 2 neurons could be replaced by 1) and do you expect existence of an analogous structure in the loss landscape ?
- In the same vein, have you considered the case where there is no nonzero local minima for a 1-subnetwork (minus 1 neurons) but there is one for a 2-subnetwork ?
- In "Best k-Layer Neural Network Approximations" by Lim et al, it is shown that for neural networks in general, the infimum might not be reachable and that minimizing the loss necessarily makes the parameter diverge. Could you comment on the relation with your work ?

**Ethical Concerns:**

["NO or VERY MINOR ethics concerns only"]

**Final Justification:**

Loss landscapes are a central object of study in deep learning. This paper offers a novel, rigorous, and thought-provoking contribution to their structure and therefore I believe should be accepted.

On top of its main contribution, which is the description of so-called "channels to infinity" and their properties it has connections to sharpness/flatness (fig 5e), mode connectivity and derivative computation by neural network (convergence to GLU and appendix C.5), which are valuable.
The use of ODE solvers, the channel-jumping technique while not central is original.
It also highlights an important discrepancy between gradient flow and gradient descent, showing that discretized optimization can become trapped at the entrance of a channel.

The paper's applicability remains limited in the context of (overparameterized or) interpolating networks. Its broader impact on practical deep learning may be constrained by the somewhat detached nature of the setting: : the focus is on non-interpolating regimes, traversing a channel requires continuous optimization, no guidance or modification of the standard training procedure is proposed and it does not offer an explanation of previously observed behaviors in deep learning. As other reviewers noted, regularization tends to truncate channel structures, and adding bias terms diminishes the effect.
Despite these limitations, which are acceptable in my view, I found the paper compelling, well-narrated and of general high quality as indicated in my original review.

**Limitations:**

yes

**Quality:**

4

**Strengths And Weaknesses:**

## Quality

The submission is high quality and seems technically sound (I have partly checked the maths) and the experiments are conducted with rigor: complete description, use of solvers when necessary and clear setup.
The paper is well narrated with theory guiding when it can a primarly experimental study.
The plots are a bit crowded but well executed and each subplot has added value.

## Clarity

The manuscript is very clear, although dense, and the organization is good as well.
The reader can find all relevant details of experiments in appendix and the authors have careful described their results.

One thing is not totally clear: I expected the channels to be sharp orthogonally to the saddle direction, as it is argued a couple of times (line 151, 203, fig 5e) but on Figure 4c it seems that the flatness converges rapidly and stays constant along the sampled direction corresponding to $e_{min}$ which is explicitly selected to have maximum negative second order variation.
If the authors could clarify why we don't see sharpening after $10^2$ that would comfort my opinion of the paper.

**Minors / suggestions**
- Figure 4d caption: $e_-$ might be a typo,  $\pm e_{min} $ would be clearer to denote a pertubation along the direction of the minimal eigenvalue.
- line 164 is not clear: $a$ is not constant so why would the $a_i$ and $a_j$ diverge ?
- the term "attractive" is mentionned a couple of times but not clearly defined. In my understanding it qualifies a minimum to which some parameter can converge but defining it in a few words and contrasting it with a stable minimum would clarify.
- the function $h$ line 176 could be specified shortly like "where $h$ is a function ..."

## Significance

Describing the loss landscape is a key for understading training dynamics and generalization.
In this regards, the paper is both impactful and interesting.
The phenomenon is actually observed on a simple setup.
Furthermore, it provides a case study of a gap between gradient flow and gradient descent as the authors report that the discretized versions can become stuck.
It might also foster the discussion on flatness in parameter space, since these channels are at the same time flat and sharp, depending on the direction.
Similarly, it sheds light on mode connectivity since channels can be composed of low loss regions that end up in a global minimum at infinity.

However, as hinted in the "Related Work" section, the applicability to the interpolating regime, which is common with overparametrized networks, is limited.
In my understanding, the phenomenon of *channel to infinity* happens in the vicinity of local minima (including global ones) of subnetworks, provided these minima do not reach 0 loss. However one can argue that with sufficient capacity, a subnetwork with one less neurons would only have global, $0$-loss minima and so channels to infinity would not be present.
Moreover, works have shown that even in mild overparametrization it seems not easy to reach local minima (e.g. "Spurious Local Minima are Common in Two-Layer ReLU Neural Networks" by Safran and Shamir) so entering a channel to infity might also be quite unlikely.
The experiments are limited to simple cases (which is fine) but it is therefore also not clear how frequent entering in a channel happens in practice: whether it happens for different kinds of architecture, nor the precise interplay with discrete optimizers and their peculiarities (edge of stability, loss catapult).

All in all the paper provides insights **and** raises new questions: it is significant.

## Originality

Loss landscape have been tremendously studied, this works uncovers a novel structure therein.
Methodologically, the use of ODE solvers and the jumping technique to overcome very slow convergence is original for the field.

---

> ### Author Rebuttal · Authors · 2025-07-31
>
> Thank you very much for your thorough review of our work, your careful assessment, and the positive evaluation. It was humbling to read that "*loss landscape have been tremendously studied, this works uncovers a novel structure therein.*" We will address your questions, concerns, and suggestions in detail below.
>
> **Sharpening of channels:**
>
> Indeed, the channels are very sharp in some orthogonal directions, and these directions become progressively sharper along the channel. This is not visible in Fig. 4c because the Hessian is evaluated at the saddle-line for a negative eigenvalue, i.e., it shows the direction which "leads away" most strongly from the saddle-line. This direction simply does not correspond to the direction of maximum curvature inside the channel in the high-dimensional parameter space. To give a more formal argument, we extended our scaling analysis to the eigenvalues of the Hessian which shows that the maximum eigenvalue diverges as $\lambda_\mathrm{max} \sim 1/\epsilon^2$.
>
> **Discrete optimizers:**
>
> This is a very helpful comment! We can directly combine the above scaling of the sharpness with the main result of the 'edge of stability': the maximum eigenvalue scales inversely with the learning rate $\eta$, i.e., $\lambda_\mathrm{max} \sim 1/\eta$. This results in a simple relation between the learning rate and the maximum depth: $\epsilon_\mathrm{min} \sim \sqrt{\eta}$. We empirically confirm in the setup of Figure 6 that gradient descent converges to the level in the channel in which $\lambda_\mathrm{max} = 2 / {\eta}$, and that the maximum eigenvalue diverges with $1/\epsilon^2$.
>
> **Vicinity of local minima:**
>
> We would like to clarify that it is *not* necessary to find local minima of smaller sub-networks in order to find channels. We sincerely apologize because this misunderstanding clearly arose from a crucial omission on our side: The experiments in Fig. 5, including the quantification of channels in 5c, all start from random initial conditions (Glorot normal). Thus, channels are found in significant number starting from random initial conditions; it is not necessary to first identify minima of sub-networks. We will add this important point by appending "*; importantly, we start in all cases from random initial conditions (Glorot normal initialization)*" to line 135. We believe this also answers your question (ii), but please let us know in case we misinterpreted this question.
>
> **Interpolating regime:**
>
> We agree that it seems unlikely to find a channel in the interpolating regime, certainly deep in the interpolating regime with large manifolds of zero-loss solutions. We will change the first sentence of the discussion (line 205) to: "*Neural network loss landscapes are non-convex, but the empirical success in training shows that their structure is benign also outside the deeply over-parameterized regime.*"
>
> **Generalization to multiple neurons:**
>
> The generalization of similar structures to multiple neurons is an interesting question. Empirically, we find channels composed of multiple neurons in various settings (see Fig 5g), which effectively makes them multi-dimensional. From a theoretical perspective we show in appendix C.5.2 how three neurons could together implement a second derivative using two divergences with different speed ($1/\epsilon$ and $1/\epsilon^2$) along two particular directions in parameter space. Empirically, we verified that a few neurons could fit to zero loss a target that is a linear combination of the activation function and its first and second order derivative.
>
> **Relation to Lim, Michalek, and Qi (2022):**
>
> Thank you very much for pointing out this interesting result, which we were unaware of. First, Lim, Michalek, and Qi explicitly construct an example of a minimum at infinity with multiple (six) data points for their Theorem 1, hence we will change line 67 to "*but there are examples, where minima at infinity can arise with two or more datapoints [Josz23,Lim22]*". However, their construction is different from a channel to infinity because only a single readout weight diverges, in contrast to the joint divergence of two readout weights that characterize a channel. Furthermore, Lim, Michalek, and Qi show for a single-hidden layer network that any sequence of weights converging to a global minimum at infinity is unbounded. This is consistent with the example in Fig. 6 where the upper channel (channel 2) leads to the global minimum, hence we will add "*; we note that this is consistent with Proposition 1 in [Lim22]*" to line 197. Importantly, channels do not only occur in combination with a global minimum at infinity: the lower channel (channel 1) in Fig. 6 provides an explicit example. Thus, channels are consistent with Proposition 1 of Lim, Michalek, and Qi but are a more generic phenomenon because they are not tied to a global minimum at infinity. Moreover, in contrast to Lim et al, our example of Fig. 6 shows that solutions at infinity can arise also with infinite data.
>
> **Minor suggestions:**
>
> Thank you very much for bringing unclear notations and wordings to our attention! (i) We will fix the notation of mineigval direction following your suggestion. (ii) Throughout section 4 we consider the limit $\epsilon \to 0$ at fixed $a$ (convergence of $a$ is shown in Fig.\ 6a, middle panel). This was indeed not clearly defined in the text and we will fix the definition of the limit to "$\epsilon\to0$ with fixed $\mathbf{w}$, $\mathbf{\Delta}$, $a$, $c$". (iii) Precisely, with 'attractive' we mean a point where gradient flow can converge, e.g., on the plateau saddle. We will add "*attracting (all eigenvalues of the Hessian are non-negative)*" to line 103.  (iv) Yes, very good suggestion; we will add "*, and $h(\mathbf{\theta}_0; \mathcal{D})$ denotes the leading order correction term*" to the end of the sentence in line 177.
>
> **Figures:**
>
> The impression that the figures are too crowded was also echoed by the other reviewers, in particular Figs. 2 and 3 seemed to obstruct the flow of the manuscript. To address this we will combine them into one figure and move the top right panels in Fig. 2 as well as panels a, b, e in Fig. 3 to the supplement. Please do let us know if you have further suggestions to improve the readability!
>
> **Conclusion**:
>
> Thank you very much again for meticulously checking and deeply engaging with our work. We sincerely appreciate your constructive criticism as well as your helpful questions and suggestions, and we hope our rebuttal was able to comfort your opinion of our work.

---

> > ### Comment · Reviewer_eBLi · 2025-08-04
> >
> > Thank you for the detailed answer and the additional insights provided.
> >
> > I have a couple of follow-up which are more speculative, please feel free to answer only to the most relevant parts.
> >
> >
> > ### Vicinity of local minima (continued)
> >
> > Yes, it was clear that the channels can be reached through optimization in the main network, and I realize my original review was ambiguous, sorry.
> > When I wrote "Moreover, works have shown that even in mild overparametrization it seems not easy to reach local minima [...], so entering a channel to infinity might also be quite unlikely.", I simply meant that local (but non-global) minima tend to disappear rapidly as the number of parameters increases.
> >
> > Let me try to clarify.
> >
> > I understood that the channels are located alongside a $\gamma$-line, which corresponds to an extension in parameter space of a minimum of a subnetwork by duplication.
> > So for each channel to infinity, there is a corresponding non interpolating minimum of a subnetwork (cf Association between channels and $\gamma$-lines from reviewer Po89). This is because an interpolating minimum could not have channels with strictly inferior loss alongside. Is this interpretation correct ?\
> > Then, if the network does not contain any subnetwork with non-interpolating minima, channels to infinity would not exist.
> > However, if the network does contain a “second-order subnetwork” (i.e., one with two fewer neurons) that admits a non-interpolating minimum, do you have any insights or intuition about whether a similar channel-like structure would exist in the loss landscape ?
> >
> > ### Catapult
> > As the channels progressively sharpen, the parameter will surely be close to an sharp unstable (cf edge of stability) region with updates slightly biased towards sending it within. Do you think there could be a catapulting mechanism out a channel during late training ?
> >
> > ### Practical guidelines
> >
> > While your work is primarily descriptive, do you envision any actionable guidelines that could emerge from your findings, particularly regarding the learning process itself?\
> > Examples:
> > - Since these channels appear to enhance the network's computational capacity, might it be beneficial to actively promote their formation ?
> > - Could their emergence serve as a signal for early stopping ? (cf Figures 5d and B11)
> > - If training with some activation function results in an unusually high number of channels, might this indicate that the network is compensating for limitations in expressivity, thereby suggesting a change in activation function ?

---

> ### Author Response · Authors · 2025-08-07
>
> Thank you very much again for your careful review, and also for actively engaging in the follow up discussion.
>
> **Vicinity of local minima**
>
> Each channel runs parallel to a $\gamma$-line. However, this $\gamma$-line could be quite far away in parameter space. Empirically, we observe that perturbations from (non-zero loss) $\gamma$-lines can lead to channels, but currently we do not know, if each channel has a higher loss $\gamma$-line nearby.
>
> It is easy to construct examples, where the global minimum is in a channel, even in large scale settings. Take, for example, an arbitrarily large network as a target network and replace two units with the gated-linear unit expression on the right-hand-side of Eq. 4: the global minimum of any student network will be at the end of a channel for infinite data (and probably at least some way into the channel for finite data), even if the student is wider than the teacher. Whether global minima are in channels also for typical machine learning datasets is an open question.
>
> Even in cases where typical initializations lead to a zero-loss solution, local minima, including channels, could exist at higher losses. We agree with your intuition that they could be found with perturbations of $\gamma$-lines that are constructed from local minima of "higher-order subnetworks".
>
> **Catapult**
>
> This is an interesting question. In the experiments with infinite data and finite learning rate, we did observe an effect that could be related: for the higher-loss channel in Fig. 6 (at $\alpha\approx-1$) the dynamics jump into the lower-loss channel (at $\alpha \approx 2.5$).
> However, this occurred only with large learning rates and only from high-loss to low-loss. Otherwise, the dynamics stop at the point where the maximum curvature in the channels reaches the edge of stability. Thus, there does not seem to be a generic catapult mechanism leading out of a channel.
> We hypothesize, however, that a jump would occur with a sudden increase of the learning rate at the point where gradient descent with a low learning rate gets stuck in the channel.
>
> **Practical Guidelines**
>
> These are very interesting questions, and we will add some of them as an outlook to the extended discussion in the revised manuscript (please let us know if you disagree with this). However, we need more empirical results with setups used in practice to be able to provide reliable guidelines. For example, we can find channels in bigger networks (VGG trained on CIFAR10; see our answers on "significance for practical scenarios" to reviewers Po89 and YsiZ) but we did not yet investigate under which circumstances going down the channels is beneficial for generalization.

---

### Official Review · Reviewer_YsiZ · 2025-06-30

**Clarity:** 3
**Significance:** 2
**Originality:** 4
**Rating:** 4
**Confidence:** 4

**Summary:**

This paper identifies a special structure in the loss landscape of neural networks: channels along which the loss decreases extremely slowly, accompanied by the divergence of output weights of at least two neurons to infinity. The work begins with an empirical investigation of the symmetry-induced saddle lines created by neuron splitting techniques. It then describes the “channel” structure observed in these networks, showing that along these channels the loss decreases slowly. The authors interpret this structure as effectively implementing a gated linear unit. Finally, they discuss the implications of this phenomenon and demonstrate how common optimizers such as SGD and Adam converge to these structures in practice.

**Questions:**

# Major Questions

- What would happen if you add weight decay to the training? Naturally, the minima of the regularized loss could not lie at infinity anymore, as at some point the weight decay would outweigh the benefit of decreased loss from following the channels to infinity. I think it’s worth expanding line 198 into a short discussion on this.

- You mentioned that along the channels there is increasing sharpness. Does it saturate at some value? Have you considered the connections to the edge of stability literature? How does the learning rate interact with how far “down” the channels the network goes?

# Minor Comments

- While reading Section 2, I found myself jumping between Figures 2 and 3. If possible, could you consider a more natural organization of the subfigures or a way to place these figures closer together and where they are referenced in the main text? Both figures have a lot of content; you might even consider merging them and moving some subfigures to the appendix.

- In Figure 3, consider using a different colormap in (b) for γ than what is used for the loss landscape in (a).

**Ethical Concerns:**

["NO or VERY MINOR ethics concerns only"]

**Final Justification:**

I think this is a very creative work with a lot of potential, but "the greatest weakness of this work is that it is unclear what the practical significance of these channels is". The authors have discussed this partially in their rebuttal, but I think they could spend more time in the discussion making this argument clear. Overall, I still recommend acceptance for this work.

**Limitations:**

yes

**Paper Formatting Concerns:**

No issues.

**Quality:**

4

**Strengths And Weaknesses:**

# Strengths

- I think this is a very creative and original study of the loss landscape of neural networks that could prove to be very insightful for understanding trained neural networks, providing insights for both future theoretical and empirical studies.

- Very strong figures that help visualize the central ideas of the work, which would otherwise be harder to follow.

# Weaknesses

- It seems that all experiments were done in very small model settings using synthetic data and often with ODE solvers rather than gradient descent. While I understand the motivation for simplified settings, I would suggest adding at least a couple of experiments using natural datasets or investigating the weight structures of pre-trained networks to strengthen this work.

- While in some sense the detailed figures are a strength, many have too many subfigures, leading to overly long captions that break up the narrative flow and make it more difficult to follow the main points of the work. I would suggest deferring some of these subfigures to the appendix.

- The greatest weakness of this work is that it is unclear what the practical significance of these channels is. While you suggest that these channels might be beneficial for generalization or could explain recent discoveries of star-shaped connectivity in loss landscapes, these arguments are purely speculative. I believe stronger evidence of these channels in more realistic networks would improve the impact of this work.

---

> ### Author Rebuttal · Authors · 2025-07-31
>
> Thank you very much for your careful reading of the manuscript, your insightful detailed, and helpful suggestions. We are pleased to read that you found our work to be "*very creative and original*" and that it "*could prove very insightful*". We address your concerns and suggestions in detail below.
>
> **Investigating pre-trained networks**:
>
> This is an interesting point, but it is difficult to address it adequately. We expect channels to infinity to be present in the loss landscape of large neural networks. Indeed, preliminary experiments applying the jump procedure (Appendix C.1) to a VGG pretrained on CIFAR10 leads into a channel. However, the problem is to identify under which circumstances the training dynamics lead into a channel --- addressing this requires significant additional work. Furthermore, the channels may not be trivial to identify, because standard training with SGD and regularization may lead only to the very beginning of it.
>
> **Edge of stability**:
>
> This is an excellent suggestion. Extending our scaling analysis (lines 180-181; Fig. 6 top row) we can show that the maximum eigenvalue of the Hessian $\lambda_\mathrm{max}$ diverges with $1/\epsilon^2$ in a channel. Thus, the sharpness does not saturate and, following the "edge of stability", any optimizer with finite learning rate will eventually get stuck. Using the scaling of the sharpness, we can quantify where the optimizer gets stuck for a given learning rate $\eta$: according to the edge of stability, the maximum eigenvalue scales inversely with the learning rate, $\lambda_\mathrm{max}\sim1/\eta$, leading to $\epsilon_\mathrm{min} \sim \sqrt{\eta}$. We empirically verified this on the setup of Fig. 6: we find indeed that GD trajectories get stuck at $\lambda_\mathrm{max}=2/\eta$. Thus, this setup presents a stylized example of the edge of stability phenomenon, where the geometrical properties of the landscape are defined by the channel.
>
> **Weight decay**:
>
> Indeed, weight decay would prevent the dynamics from continuing along the channel. Nonetheless, the computation would be a finite-difference approximation of the directional derivative that gives rise to the Gated Linear Unit, which provides a clear understanding of the underlying computation. We will add the following brief discussion to the paragraph in lines 223-227 to expand the brief statement from line 198: "*Adding regularization has a similar effect: from a certain point the regularization outweighs the decrease in loss such that the dynamics get stuck inside the channel. Nonetheless the network implements an approximation of the gated linear unit.*"
>
> **Figures 2 \& 3**:
>
> This is a very helpful suggestion. We will follow your advise and combine parts of Figs. 2 and 3 into a single figure. To this end, we will move the top right panels of Fig. 2 and panels a, b, e from Fig. 3 into the appendix. Fig. 3 c, d will be combined with the remainder of Fig. 2, thereby summarizing the main results on saddle-lines in a single figure.
>
> **Conclusion**
>
> Thank you very much again for the work that you put into this review and the helpful comments. We are particularly grateful for your suggestion of the link to the "edge of stability", which we believe provides an interesting link to the literature and improves the significance of our results. If you agree, we would appreciate if you would revise your scores to reflect this.

---

> > ### Comment · Reviewer_YsiZ · 2025-08-05
> >
> > Thank you for your response.
> >
> > Your discussion of edge of stability in fig. 6 is interesting and I look forward to seeing this explained in the updated manuscript.
> >
> > Overall, I think this is a very creative work with a lot of potential, but I still agree with my original review that "the greatest weakness of this work is that it is unclear what the practical significance of these channels is". Your reply hasn't addressed this point.
> >
> > I will thus keep my original evaluation of 4, recommending acceptance, but I encourage the authors to use the additional space in the updated manuscript to expand on your discussion of:
> > - how these channels might be beneficial for generalization
> > - why you "expect channels to infinity to be present in the loss landscape of large neural networks" and your attempts/challenges to find them empirically

---

> > > ### Author Response · Authors · 2025-08-07
> > >
> > > We thank you very much again for critically engaging with our work and your helpful comments and suggestions. As we pointed out in our response we can find channels in the loss landscape of large networks (in this case VGG trained on CIFAR10); we will discuss these results in the updated manuscript. A simple example where the channels are clearly beneficial for generalization is a teacher-student scenario where the student needs a directional derivative, i.e., a channel, to implement the teacher function; we will also add this to the discussion. However, we believe it needs carefully designed follow up experiments to obtain reliable insights into the relation between channels and generalization in more generic scenarios, hence we will only briefly speculate about this in the extended discussion.

---

### Official Review · Reviewer_Po89 · 2025-07-02

**Clarity:** 4
**Significance:** 3
**Originality:** 3
**Rating:** 5
**Confidence:** 4

**Summary:**

This paper discusses a particular type of structure that the authors have discovered in the loss landscape of neural networks with dense layers, that the authors call channels to infinity. These structure indeed appear as narrow valleys ("channels") with a very slow downward slope along a single line that points to a minimum at infinity. The authors offer an explanation of how these channels may arise in the context of an overparameterized dense layer and examine their property in the infinite limit (they show that two perceptron-like units behave effictvely as a perceptron unit plus a gated linear unit). They perform numerical experiments to look for such structures in networks trained on relatively simple tasks, finding them to be rather common.

**Questions:**

* Although the treatment is general with respect to the activation function, the ones that is used by the authors is very unusual, softplus(x)+sigmoid(4x). Thus, in the empirical estimation of the relevance of the phenomenon, it is unclear to me how much the results would change by using a more common activation function. Indeed, the authors themselves acknowledge that (line 220): "It is also an open question, to what extent channels to infinity arise in MLPs with the popular, non-smooth ReLU activation function." I guess that the authors may have chosen not to use ReLUs in order to get rid of extra symmetries that affect those networks, as these would create extra flat directions (one per hidden unit). On top of this, because of the symmetry, the norm of each unit is no longer well-defined in a sense, since the norm of each unit can be changed arbitrarily by using a transformation of the kind $ReLU(\alpha x) = \alpha ReLU(x)$ for any $\alpha > 0$. In such scenario, defining what would qualify as a "channel to infinity" as described in this paper is not even clear, I suppose.
* Besides the activation function, the size of the network and the complexity of the training task may also play a significant role.
* Maybe I missed it, but is it trivial that a channel to infinity is always associated to a $\gamma$ line? From the discussion in the appendix it seems like identifying such lines is problematic.
* The authors claim that SGD and ADAM tend to get to a channel to infinity and get stuck there (line 201). Wouldn't momentum be able to fix this entirely? (Also possibly some modification of ADAM that worked in a rotated parameter space, so as to adapt the gradient step and increase it in the flat direction, but that's way more speculative)
* Figure 5c shows that (purported) channel and non-channel minima have similar training loss. But what about the test loss, or test accuracy? Any difference in that regard? How does the test accuracy change along the line, is it also affected only slightly?
* The procedure for identifying the channels to infinity in fig. 5 is based on thresholding the distance between pairs of hidden units and their norms (5a). This seems to be a necessary but not sufficient condition. Then, there is an estimate of how many of these are associated to lines that go parallel to some $\gamma$ line (figures 5d and B9 and B11). The latter seems to identify at most 60% (depending on the dataset and other details) of networks for which this is the case. So if I understand correctly a more conservative estimate of the number of these channel to infinity minima should be scaled by 60% or less. Or am I missing something?
* The suggestion that channels to infinity may explain the observed star-shaped connectivity in loss landscapes (line 228) is unconvincing to me in light of the results found in this paper https://arxiv.org/abs/2202.03038 where a star-like (or "octopus-shaped", as the authors put it) structure was observed in a network with removed symmetries where all neurons were normalized; and also in this paper https://arxiv.org/pdf/2305.10623 where a star-shape structure was shown theoretically to exist in a simple non-convex model, in that case with a single hidden unit.

**Ethical Concerns:**

["NO or VERY MINOR ethics concerns only"]

**Final Justification:**

After the rebuttal all my concerns were adequately addressed. Having seen the other reviews, there seems to be a general consensus. Therefore, my initial doubts were solved (I'm increasing my significance score from 2 to 3) and I'm now recommending acceptance (I'm increasing the rating from 3 to 5).

**Limitations:**

yes

**Quality:**

3

**Strengths And Weaknesses:**

Strenghts:
* Clearly written, reproducible, appendices complete, overall very well presented
* Interesting addition to the body of work on the geometry of the loss landscape. At the very least, these finding suggest that the kind of structures revealed here should be taken into account in the discussions and analyses of the loss landscape. Possibly, more of these kinds of structures may be uncovered in the future.

Weaknesses:
* Despite my last comment above, to me the significance of this contribution remains unclear, mostly because:
  - I'm not sure that the evidence is sufficient to actually claim this to be a relevant phenomenon in practical scenarios (more on this in the "questions" box)
  - It's also unclear if and how these structures affect the learning process and the generalization properties of the network
  - The fact that the addition of biases drastically reduces the effect (as shown in Appendix A) suggests that, indeed, the phenomenon may "go away" in more realistic scenarios
  - I also have some significant doubts about the proposal (made in the discussion) that these kind of structures may explain the star-shaped geometry of the space of solutions observed empirically in other works

---

> ### Author Rebuttal · Authors · 2025-07-31
>
> We thank you very much for your detailed reading of our manuscript, your insightful questions and suggestions, and the balanced evaluation. We truly appreciate your effort put into reviewing our work and we are pleased to read that you found it "*interesting*" and "*very well presented*". We address your questions and concerns in detail below. In particular, we would like to clarify two important points regarding the activation function and the effect of biases that might have biased your rating.
>
> **Effect of adding biases**:
>
> Adding biases does *not* have a negative impact on the number of channels, all experiments in Fig. 5b are with biases. In contrast, adding biases strongly affects the presence of plateau-saddles (local minima on $\gamma$-lines) -- they indeed seem to "go away" in more realistic scenarios, as expected from inspection of the structure of the Hessian matrix (see lines 113-114 and supplemental Fig. A1 and our response to reviewer `jaHr` on this point).
> Thus, simply adding biases does not suggest that the channels go away in more realistic scenarios, in contrast, plateau-saddles disappear.
>
> **Unusual activation function**:
>
> This comment seems to be based on a misunderstanding. We used a range of different activation functions throughout the manuscript: Figs. 2, 3 are indeed based on $\mathrm{softplus}(x)+\mathrm{sigmoid}(4x)$ but Figs. 4, 5 are based on $\mathrm{softplus}(x)$ (see appendix B3; a preliminary analysis for $\mathrm{erf}$ is consistent with the $\mathrm{softplus}$ results), and Fig. 6 is based on $\mathrm{erf}(x)$ (see line 183). Thus, our results show that channels to infinity occur with different activation functions.
>
> **Significance for practical scenarios**:
>
> We expect channels to infinity to be present in the loss landscape of large neural networks. Indeed, preliminary experiments applying the jump procedure (Appendix C.1) to a VGG pretrained on CIFAR10 leads into a channel. However, the tricky question is under which circumstances the training dynamics lead into a channel --- addressing this requires significant additional work. Furthermore, the channels may not be trivial to identify, because standard training with SGD and regularization may lead only to the very beginning of it.
>
> **Generalization**:
>
> Train and test loss are almost identical in our setups because the data is relatively low-dimensional and densely sampled. In particular, the distribution of the test loss for channels and finite minima is similarly overlapping as the distribution of the training loss.
>
> **Momentum**:
>
> The reason why SGD and ADAM get stuck is their finite step size in combination with the increasing sharpness (maximum eigenvalue of the Hessian) of the channels. As shown in the literature on the "edge of stability" [Cohen, et al.; Gradient Descent on Neural Networks Typically Occurs at the Edge of Stability], the maximum sharpness is inversely related to the step size $\eta$. Note that this effect occurs with and without momentum (see their Eq. 1). Extending our scaling analysis (lines 180-181; Fig. 6 top row) we can show that the sharpness diverges with $1/\epsilon^2$ in the channels.
> Furthermore, SGD training in the setup of Figure 6 converges exactly at a point in the channel corresponding to the edge of stability ($\lambda_\mathrm{max} = 2/\eta$).
> Regarding your proposal of an optimizer operating in "a rotated parameter space", we believe this is very similar in spirit to the jump procedure that we used to proceed when the ODE solver got stuck (lines 169-172; appendix C.1).
>
> **ReLU activation**:
>
> We avoided the ReLU activation due to its non-smoothness (the homogeneity can be accounted for, see next point). ReLU loss landscapes present non-differentiable cusps at points in parameter space defining the boundary between a ReLU neuron being activated or dead for a specific datapoint.
> We noticed, especially nearby saddles where trajectories can get very close to each other, that ReLU network trajectories would get stuck at cusp-like points near the $\gamma$-line.
> However, given that softplus is a smooth approximation of ReLU, we speculate that ReLU landscapes can still present channel structures.
>
> **Identifying channels**:
>
> This is an excellent point and we thank you very much for raising it. Indeed the threshold on weight similarity provides only a necessary condition.
> Thus, we will scale the numbers reported in Fig. 5b down by including only trajectories that have > 90\% fraction of the updates parallel to the saddle line. This reduces the number of confidently identified channels to 80-90\% the reported number (depending on the dataset), which is still a significant fraction of runs.
> On top of this, we improved the channel identification criteria in two important ways:
> 1) We now account for the approximate homogeneity of softplus (in the limit of large inputs ReLU and softplus coincide) which allows to rescale input- vs. output-weights to correctly identify putative channels with high norm input weights.
> Specifically, channels of high input weight norm softplus neurons can have weights grow both parallel to the saddle line *and* parallel to the direction of the input weight vector (becoming effectively a two dimensional space).
> 2) The originally selected threshold for the rosenbrock data was including solutions that were not channels (see dashed line in Fig B8). We selected better thresholds for each dataset and saw a drastic reduction of runs that had updates not parallel to the saddle line (or the extra manifold described above). The down-scaling by 80-90\% already includes these two corrections.
>
> **Association between channels and $\gamma$-lines**:
>
> Quite often, a small perturbation from a $\gamma$-line leads to a channel. However, it is an open question, whether the attractive region of all channels contains points that are arbitrarily close to a $\gamma$-line. In some preliminary work, we tried with mixed success to find $\gamma$-lines nearby the region of attraction of given channels. A direct association between $\gamma$-lines and channels is that they are asymptotically parallel. To see this we can rewrite the reparameterization introduced in section 4 as $\mathbf{\theta} = (\mathbf{w_1}, \ldots, \mathbf{w}, \mathbf{w}, a_1, \ldots, c/2, c/2) + (\mathbf{0}, \ldots, \epsilon\mathbf{\Delta}, -\epsilon\mathbf{\Delta}, 0, \ldots, a/2\epsilon, -a/2\epsilon)$ where the update given by the second term is asymptotically equal to $\frac{a}{2\epsilon}(\mathbf{0}, \ldots, 0, 1, -1)$, which is parallel to a $\gamma$-line.
>
> **Star-shaped connectivity**:
>
> We would like emphasize that the connection to star-shaped connectivity was clearly indicated as a speculation (line 228). That being said, we agree that the connection is a bit premature since channels could only explain aspects of star-shaped connectivity---clearly, it could not explain the structure observed by Annesi et al. for a spherical perceptron (but note that they consider geodesics on the sphere, not in euclidean space)---and, re-reading our formulation from this point of view, we agree that the link to star-shaped connectivity is premature. We will remove this speculation from the discussion.
>
> **Conclusion**
>
> We thank you again for the careful reading and the insightful questions and comments. We hope our rebuttal convinced you that the channels can be of practical significance; in particular we hope we were able to clarify that the channels occur for commonly used activation functions and that they also occur with biases. If this is the case please consider raising your overall score to reflect this.

---

> > ### Comment · Reviewer_Po89 · 2025-08-04
> >
> > Thank you very much for your replies and clarifications. My initial concerns were addressed and I'm satisfied with the proposed changes; I'll thus increase my score to 5.
> >
> > There is still a minor thing I'd like to ask though, in one of the responses you wrote "We avoided the ReLU activation due to its non-smoothness (the homogeneity can be accounted for, see next point)" but I don't see the part of how to account for homogeneity in the rest of the rebuttal, I'd be curious to see it.

---

> > > ### Author Response · Authors · 2025-08-07
> > >
> > > Thank you very much for reconsidering your score! We meant that we can treat the homogeneity of the ReLU in the same way as we treat the approximate homogeneity of the softplus function (point 1 in "Identifying channels" in the rebuttal). We apologize for the short and compressed explanation.
> > >
> > > To identify channels in ReLU networks, we account for the positive homogeneity of the nonlinearity, $a\,\mathrm{ReLU}(\boldsymbol{w}\cdot\boldsymbol{x}) = \frac as\,\mathrm{ReLU}(s\,\boldsymbol{w}\cdot\boldsymbol{x})$ for $s>0$, by setting $s=1/\|\boldsymbol{w}\|$ which constrains the (scaled) input weight norm to one.
> > > In the rescaled parameters we keep track of the cosine similarity of the input weights $\cos(s_i\boldsymbol{w}_1, s_j\boldsymbol{w}_j)$, the absolute value of the readout weights $\left|\frac{a_i}{s_i}\right| + \left|\frac{a_j}{s_j}\right|$, and the cosine similarity of a $\gamma$-line with the update of the rescaled parameters.
> > > In addition to the structure of the channels described in the paper, ReLU channels contain thus an equal-loss manifold defined by the positive homogeneity --- note that this is a generic geometrical structure in the loss-landscape of ReLU networks, irrespective of being in a channel or not.
> > > It could be that gradient flow moves both in direction of the channel and some direction in the homogeneity manifold; we saw this behavior explicitly in the case of softplus channels with high norm input weights.
> > > Furthermore, we just ran a toy example with infinite data (akin to Fig. 6) in 2 dimensions with target function $x_2  \cdot \mathrm{heaviside}(x_1)$ and a network with 2 ReLU neurons: training runs indeed lead into a channel, with both the norm of the input weights and the output weights growing simultaneously.

---

### Official Review · Reviewer_jaHr · 2025-07-02

**Clarity:** 3
**Significance:** 4
**Originality:** 3
**Rating:** 5
**Confidence:** 2

**Summary:**

This paper explores the loss landscape around $\gamma$-lines.   A $\gamma$-line is created when a small network is trained to a local minimum and then one of the nodes is cloned, creating a line in parameter space of networks computing the same function.  The paper's experiments show that $\gamma$-lines arise naturally in the training process.  Extensive experiments on small networks shed insight into the loss landscape around these $\gamma$-lines and proved evidence that they are accompanied by parallel nearly-flat channels that lead to local optima at infinity.  The paper also shows that, in a limit,  these channels mimic a kind of gated linear gate.

**Questions:**

Some minor questions.
In figure 2, it appears that different numbers of runs were used for the different network sizes.  Could this help explain the different numbers of unique minima discovered?

Figure 4D shows that updates eventually become essentially parallel to the $\gamma$-line, does the distance from the $\gamma$ line remain small?

Why might $\gamma$-lines lose plateau-saddles when biases are added?

**Ethical Concerns:**

["NO or VERY MINOR ethics concerns only"]

**Final Justification:**

My initial review mostly stands: this appears to be a solid contribution to the understanding of neural net loss landscapes.
The author's comments gave me a greater appreciation for the possible value of the connection to gated linear units, but
I still feel that the other aspects of the paper are stronger, but now see how others more enthusiasm for the channel phenomenon's relationship to gated linear units.

**Limitations:**

The behavior explored is in the absence of regularization.  Any resonable regularization would seem to prevent to formation of \gamma$-lines and channels.

**Quality:**

3

**Strengths And Weaknesses:**

The channel structure uncovered by the paper is interesting, and the experimental setup (although on toy networks) is reasonable.
The experiments convincingly demonstrate that $\gamma$-lines can arise naturally, and that the associated channels are likely to have
the nearly-flat but converge at infinity behavior.  It seems to be a valuable contribution to the study of loss landscapes in un-regularized networks.  The significance of the connection to gated linear units is unclear.

Although the paper is generally well written, Figure 3a was confusing to me.

---

> ### Author Rebuttal · Authors · 2025-07-31
>
> Thank you for carefully reading our paper, the fair and positive evaluation, and your questions and suggestions. We are pleased to read that our work "*seems to be a valuable contribution to the study of loss landscapes*". We address your concerns in detail below. If our answers will increase your confidence in your judgment we would appreciate if you could raise your score to reflect this.
>
> **Significance of the connection to Gated Linear Units**:
>
> When we saw the channels for the first time, and realized that they occur in a significant number of cases, we naturally wondered about their functional role. The connection to Gated Linear Units answers this question: the channels implement a new function that is the derivative of the activation function multiplied by a linear projection of the input. When naively looking at the architecture of an MLP, fitting such functions does not seem possible. As a concrete example, a network with a *finite* number of softplus neurons can perfectly fit targets composed of Gated Linear Units. Fig. 6B (left panel) provides another stylized example.
>
> **Regularization**:
>
> Indeed, regularization would prevent the dynamics to reach the "end" of the channel. However, even if the dynamics "get stuck" in the channel, the network would still employ a finite-difference approximation of the Gated Linear Unit, which provides a clear understanding of the underlying computation (see lines 226-227).
> In other words, the landscape would contain a channel whose "end" is at finite values due to regularization.
> We will expand the discussion of regularization (as also suggested by Reviewer `YsiZ`) to make sure it is clear that the idealized behavior (convergence to GLUs) can only occur in the absence of regularization. Concretely, we will add this brief discussion to the end of the paragraph in lines 223-227: "*Adding regularization has a similar effect: from a certain point the regularization outweighs the decrease in loss such that the dynamics get stuck inside the channel. Nonetheless the network implements an approximation of the gated linear unit.*"
>
> **Number of runs vs. number of unique minima in Fig. 2**:
>
> We ran $50 \cdot 2^r$ simulations per network, where $r$ is the hidden layer size. We experimentally verified that conducting more simulations for the smaller networks does not lead to the discovery of new minima. Furthermore, we have empirically noticed that the number of unique minima follows roughly $2^r$, and we estimated that running 50 times that number of simulations would allow good coverage of the landscape. There may be minima whose basins of attraction are too small or unreachable from standard Glorot initializations; those would not be quantified in our figure.
>
> **Distance between channels and $\gamma$-line**:
>
> The distance between the channel and the $\gamma$-line does not necessarily remain small because the other parameters can move significantly; this is quantified in supplemental Fig. B5.
>
> **Effect of biases on plateau-saddles**:
>
> The stability of the $\gamma$-line is dependent on two sub-matrices that can be derived from the Hessian of the loss (Fukumizu \& Amari, 2000); in Appendix A.1 we briefly describe their structure. For a plateau-saddle, these matrices must be positive or negative semi-definite (eigenvalues must be all positive or all negative), or zero. In general it is very difficult to make predictions on the signs of the eigenvalues because they depend on the whole training dataset. For the case of multi-dimensional output weights, Petzka \& Sminchisescu 2021 (section 5.5) simply note that constraints on all eigenvalues are too strong to be satisfied by any practical example with noticeable probability. We speculate that the same accumulation of constraints can explain our results for adding biases to the network. Our empirical evidence, in combination with Petzka \& Sminchisescu 2021, shows that symmetry-induced $\gamma$-lines are not a source of local minima in loss landscapes. Note that this only applies to plateau-saddles, not the channels to infinity.
>
> **Fig. 3a**:
>
> For the final version we will combine Figs. 2 and 3 to improve the flow of the manuscript (see also the comment by Reviewer `YsiZ`), which includes dropping 3a, because the main intuition that the plateau-saddle looses stability is also conveyed in 3c.
>
> Thank you very much again for your constructive feedback!

---

### Decision · Program_Chairs · 2025-09-17

**Decision:**

Accept (poster)

**Comment:**

This paper identifies a special structure in the loss landscape of neural networks: channels along which the loss decreases extremely slowly, accompanied by the divergence of output weights of at least two neurons to infinity.  The authors interpret this structure as effectively implementing a gated linear unit.  They discuss the implications of this phenomenon and demonstrate how common optimizers such as SGD and Adam converge to these structures in practice.

Reviewers praised the interesting nature of the main concept, the quality of presentation including the figures, and reproducibility.

The had reservations about the practical significance, the impact of biases, the connections to other phenomena, and the limited experimental setup.

The authors handled the rebuttal and the subsequent discussion well, and some of the reviewers increased their ratings.

Overall, this is a clear accept, with all four reviewers being in favor of it, and only one of them weakly.  Nevertheless, the authors should address all the feedback received when revising the paper.